# TSC2 regulates lysosome biogenesis via a non-canonical RAGC and TFEB-dependent mechanism

Nicola Alesi[1], Elie W. Akl[1], Damir Khabibullin[1], Heng-Jia Liu [1], Anna S. Nidhiry[1], Emma R. Garner[2], Harilaos Filippakis [1], Hilaire C. Lam[1], Wei Shi [3], Srinivas R. Viswanathan [2], Manrico Morroni[4,5], Shawn M. Ferguson [6,7] & Elizabeth P. Henske[1✉]

Tuberous Sclerosis Complex (TSC) is caused by *TSC1* or *TSC2* mutations, resulting in hyperactivation of the mechanistic target of rapamycin complex 1 (mTORC1). Transcription factor EB (TFEB), a master regulator of lysosome biogenesis, is negatively regulated by mTORC1 through a RAG GTPase-dependent phosphorylation. Here we show that lysosomal biogenesis is increased in TSC-associated renal tumors, pulmonary lymphangioleiomyomatosis, kidneys from Tsc2[+/−] mice, and *TSC1/2*-deficient cells via a TFEB-dependent mechanism. Interestingly, in *TSC1/2*-deficient cells, TFEB is hypo-phosphorylated at mTORC1-dependent sites, indicating that mTORC1 is unable to phosphorylate TFEB in the absence of the TSC1/2 complex. Importantly, overexpression of folliculin (FLCN), a GTPase activating protein for RAGC, increases TFEB phosphorylation at the mTORC1 sites in *TSC2*-deficient cells. Overexpression of constitutively active RAGC is sufficient to relocalize TFEB to the cytoplasm. These findings establish the TSC proteins as critical regulators of lysosomal biogenesis via TFEB and RAGC and identify TFEB as a driver of the proliferation of *TSC2*-deficient cells.

[1] Pulmonary and Critical Care Medicine, Department of Medicine, Brigham and Women's Hospital, Harvard Medical School, Boston, MA, USA. [2] Department of Medical Oncology, Dana–Farber Cancer Institute, Harvard Medical School, Boston, MA, USA. [3] Department of Surgery, Children's Hospital Los Angeles, Keck School of Medicine, University of Southern California, Los Angeles, CA, USA. [4] Department of Experimental and Clinical Medicine, Section of Neuroscience and Cell Biology, School of Medicine, Università Politecnica delle Marche, Ancona, Italy. [5] Electron Microscopy Unit, Azienda Ospedaliero-Universitaria, Ancona, Italy. [6] Department of Cell Biology, Yale University School of Medicine, New Haven, CT, USA. [7] Department of Neuroscience, Program in Cellular Neuroscience, Neurodegeneration and Repair, Yale University School of Medicine, New Haven, CT, USA. ✉email: EHENSKE@BWH.HARVARD.EDU

Tuberous sclerosis complex (TSC) is caused by mutational inactivation of the *TSC1* or *TSC2* tumor suppressor genes[1]. TSC affects multiple organs including the brain, heart, kidney, lung and skin[2]. While the majority of tumors in TSC are benign, malignant tumors also occur, particularly in the kidney[3]. TSC1 and TSC2 are part of the protein complex that integrates signals from the extracellular environment (oxygen, energy, nutrients, growth factors) to regulate the kinase activity of mechanistic/mammalian target of rapamycin complex 1 (mTORC1) via the small GTPase Ras homolog enriched in brain (RHEB)[4–8]. RHEB-dependent mTORC1 activation takes place on the surface of the lysosome[9]. The hallmark of *TSC1* and *TSC2*-deficient cells (hereafter referred to as *TSC*-deficient cells) is therefore hyperactivation of mTORC1, which is believed to be the primary driver of tumorigenesis in TSC.

TFEB (transcription factor EB) and the other members of the MiTF family of transcription factors (MITF, TFE3, TFEC) are master regulators of lysosomal gene expression, lysosomal biogenesis, and autophagy[10–14]. The localization and function of TFEB are tightly regulated by mTORC1 kinase activity, with phosphorylation of TFEB resulting in cytoplasmic sequestration by 14-3-3 proteins[13,15–17].

Recruitment of mTORC1 to lysosomes occurs via interactions with the Ras-related GTP-binding proteins (RAG GTPases), which are activated when amino acids are abundant[18]. In order to be phosphorylated by mTORC1, TFEB needs also to be recruited to lysosomes by the RAG GTPases[19]. Folliculin (FLCN) is a GTPase activating protein for RAGC/D[19–22] and thereby regulates both mTORC1 and TFEB abundance at the lysosomal surface[19,23,24]. Germline mutations in *FLCN* cause the hereditary cancer syndrome Birt-Hogg-Dube (BHD)[25,26], which shares some clinical phenotypes with TSC including benign skin tumors, cystic lung disease, and renal cell carcinoma (RCC).

Because mTORC1 is hyperactive in TSC, TFEB should be predominantly cytoplasmic in *TSC*-deficient cells. However, prior studies in *TSC*-deficient cells have revealed conflicting results in terms of TFEB localization[13,15,24,27–29]. Here we demonstrate that despite high mTORC1 activity in *TSC*-deficient cells, TFEB is predominantly nuclear and unexpectedly hypo-phosphorylated at the mTORC1-dependent sites. Furthermore, TFEB drives lysosomal gene expression and promotes proliferation in vitro and in vivo in *TSC2*-deficient cells. It has recently been discovered that renal tumorigenesis in BHD is TFEB-dependent[24]. This supports the concept that the regulation of TFEB is the critical mechanistic link between tumorigenesis in TSC and BHD, diseases in which there is some clinical similarity, and further highlights the possibility that TFEB may be a primary driver of tumorigenesis in TSC. Taken together, our findings indicate that TFEB is a critical disease-relevant target of the TSC proteins.

## Results

**Lysosome abundance, lysosomal gene expression, and protein levels are increased in TSC.** Renal disease is a major source of morbidity and mortality in TSC[30,31]. To elucidate the pathogenesis of renal lesions in TSC, transmission electron microscopy (TEM) was performed on the kidneys of 18-months old *Tsc2*[+/−] mice, which develop renal cysts and cystadenomas[32]. This revealed a 3-fold increase in lysosome number within cystic epithelial cells compared to normal adjacent kidney (Fig. 1a, b). Consistent with this increased lysosomal number, expression of the lysosomal cholesterol transporter Niemann-Pick C1 (NPC1)[33], was enriched in the mouse kidney cystic epithelium (Fig. 1c). Interestingly, NPC1 is also enriched in renal lesions from TSC patients (angiomyolipomas and RCCs) compared with adjacent normal kidney (Fig. 1d–f, Supplementary Fig. 1a). In

*Tsc1*[−/−] and *Tsc2*[−/−] mouse embryonic fibroblasts (MEFs), mRNA levels of lysosomal genes such as *Npc1, Niemann-pick c2 (Npc2), Cathepsin K (Ctsk),* and *Hexosaminidase (Hexa)* were increased 2- to 6-fold compared to controls (Fig. 1g, h), and protein levels of NPC1 and Cathepsin K were higher in *Tsc2*[−/−] MEFs compared with controls (Fig. 1i). These results establish that lysosomes, which are increasingly recognized as critical drivers of tumorigenesis[34–36], are enriched in TSC.

**TSC-deficiency leads to increased nuclear localization and activity of TFEB with decreased TFEB phosphorylation at Serine 142 and 211.** To understand the mechanisms driving high lysosome abundance in TSC, we focused on the MiTF family of transcription factors (TFEB, TFE3, MITF, TFEC), known regulators of lysosome biogenesis[10,11,13]. We found that *Tfeb*, but not *Tfe3* or *Mitf*, was upregulated in *Tsc*-deficient MEFs (Supplementary Fig. 1b). *Tfec* was not expressed. As noted earlier, although phosphorylation of TFEB by mTORC1 is known to result in its cytoplasmic sequestration[13,15–17], the localization of TFEB in *TSC*-deficient cells is controversial[15,16,24,27,29]. We found that TFEB is predominantly nuclear in Human Melanoma Black-45 (HMB45) and NPC1-positive lymphangioleiomyomatosis (LAM) nodules, the pulmonary manifestation of TSC[37] (Supplementary Fig. 1c–e). TFEB is also primarily nuclear in TSC-associated RCC and renal angiomyolipomas (Fig. 2a–c), and in *Tsc1/2*-null MEFs (Supplementary Fig. 2a).

To understand the mechanisms through which TFEB is nuclear in TSC, despite high mTORC1 activity, we used HeLa cells with stable expression of TFEB-GFP[13,15–17]. First, we confirmed the nuclear enrichment of exogenous GFP-tagged TFEB after either *TSC1* or *TSC2* knockdown by siRNA (Fig. 2d, e). Next, we examined TFEB's phosphorylation at the mTORC1-dependent sites (S142 and S211). Surprisingly, both mTORC1-dependent sites were hypophosphorylated in *TSC*-deficient HeLa-TFEB-GFP cells (Fig. 2f, g).

To determine if the nuclear TFEB in *TSC2*-deficient cells is active, we generated a novel luciferase construct derived from the promoter of the transmembrane glycoprotein NMB (GPNMB), a TFE3-TFEB target, and lysosomal glycoprotein[38]. *TSC2* down-regulation by siRNA increased the activity of the *GPNMB* promoter by 10-fold in HeLa cells and 27-fold in HeLa-TFEB-GFP cells (Fig. 2h). Downregulation of *FLCN*, used as positive control, increased the promoter activity of *GPNMB* by 7-fold in HeLa cells and 10-fold in HeLa-TFEB-GFP cells (Fig. 2h).

In *Tsc1*-null and *Tsc2*-null MEFs we observed increased *Tfeb* mRNA expression (Supplementary Fig. 1b), thereby in these cell lines increased nuclear localization and activity of TFEB could be due to its higher total levels. In HEK293T cells after *TSC2* knockdown by siRNA we found only a minor increase in *TFEB* mRNA expression (20%) (Fig. 3a), no appreciable change at the protein level (Fig. 3b), and primarily nuclear TFEB (Fig. 3c). Similarly, in HeLa cells after *TSC2* knockout by CRISPR we observed a 50% increase in the mRNA expression of *TFEB* (Fig. 3d), no appreciable change at the protein level (Fig. 3e), and a clear increase in nuclear TFEB after nuclear/cytoplasmic fractionation (Fig. 3f), indicating that increased nuclear localization of TFEB is not driven by increased expression.

TFE3 was also primarily nuclear in *TSC*-deficient cells (Fig. 3c, f, Supplementary Fig. 2b, c) behaving similarly to TFEB, TFE3 mRNA, and protein expression was not affected by *TSC2* loss. Overexpression of TFEB with serine to alanine mutation at the mTORC1-dependent phosphorylation sites (S142, S211, and S142/S211) resulted in nuclear localization of TFEB in both *TSC2*-expressing and *TSC2*-deficient HeLa cells (Supplementary Fig. 3).

To confirm that increased GPNMB promoter activity and increased lysosomal gene expression are TFEB-dependent in

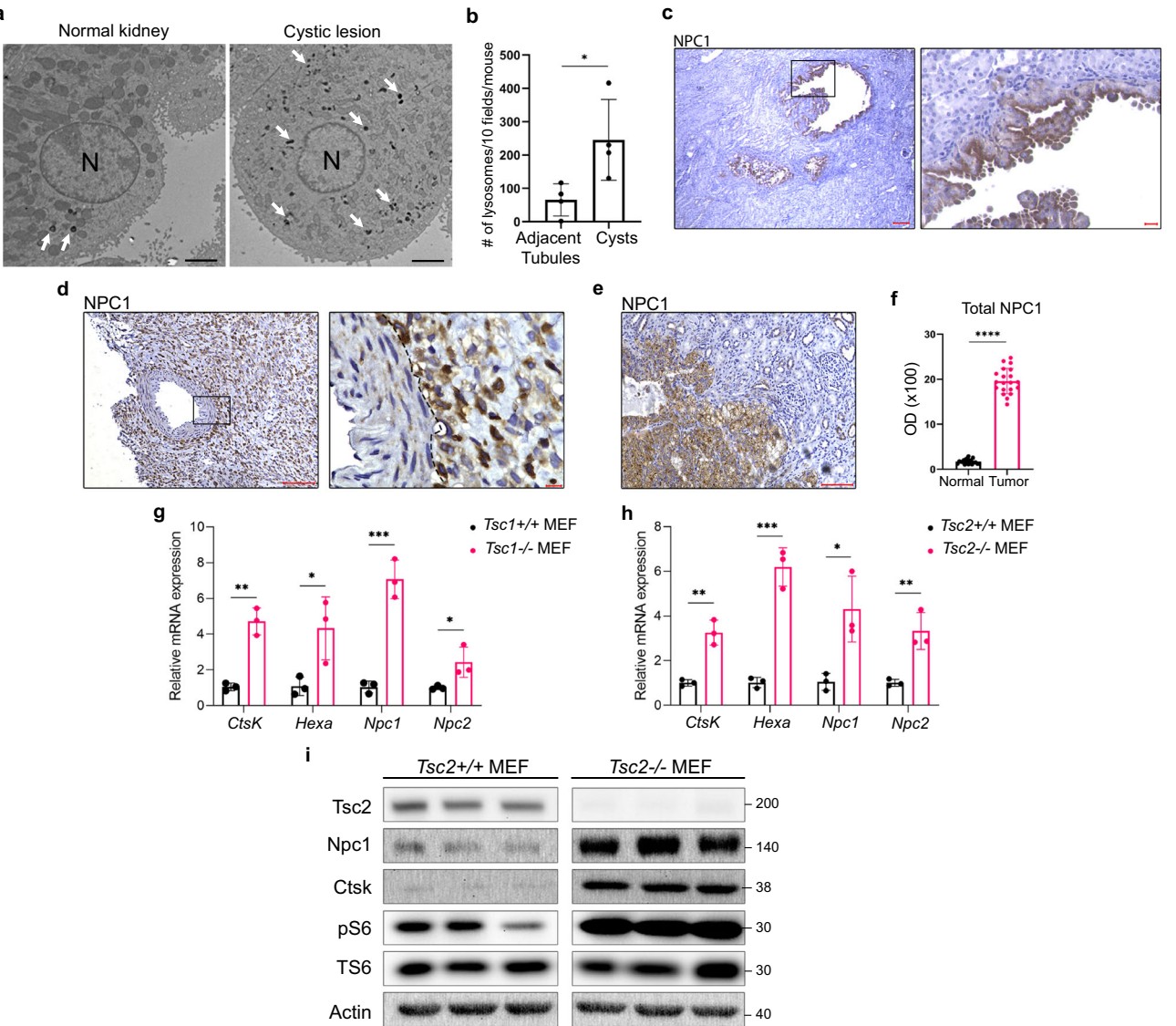

**Fig. 1 Increased lysosome biogenesis in TSC. a** Transmission electron microscopy representative images showing increased lysosomes (white arrows) in renal cyst-lining cells in 18mo $Tsc2^{+/-}$ mice compared to the normal kidney. N denotes nucleus. Scale bar = 2 μm. **b** Quantification of lysosomes in 10 fields/kidney in 18mo $Tsc2^{+/-}$ mice cysts and adjacent normal tubules ($n = 4$ kidneys), $p = 0.0285$. **c–e** Immunohistochemistry for the lysosomal marker NPC1 in renal cysts from $Tsc2^{+/-}$ mice ($n = 6$ kidneys), (left image scale bar = 100 μm, right image scale bar = 20 μm) (**c**), human renal angiomyolipoma ($n = 3$ patient samples) (left image scale bar = 100 μm, right image scale bar = 10 μm) (**d**), and TSC-associated renal cell carcinoma ($n = 3$ patient samples). Scale bar = 100 μm (**e**). The dashed line in d shows the boundary between angiomyolipoma cells on the right and a blood vessel on the left. **f** NPC1 optical density quantified for TSC-associated renal cell carcinomas (20 measurements on 5 random areas of tumor and normal adjacent kidney quantified per section in 3 patient samples). **g, h** qRT-PCR analysis of lysosomal genes in $Tsc1^{+/+}$ and $Tsc1^{-/-}$ MEFs (**g**) and $Tsc2^{+/+}$ and $Tsc2^{-/-}$ MEFs (**h**), ($n = 3$ biological replicates per condition). **i** Immunoblot analysis of the lysosomal proteins NPC1 and Cathepsin K (CTSK) in $Tsc2^{+/+}$ and $Tsc2^{-/-}$ MEFs ($n = 3$ biological replicates per condition, samples not contiguous, from the same gel). Graphs are presented as mean ± SD. Statistical analysis in b was performed using the Mann–Whitney $U$ test, $*p < 0.05$. Statistical analyses in (**f**), (**g**), and (**h**) were performed using two-tailed Students $t$-test, $*p < 0.05$, $**p < 0.01$, $***p < 0.001$, $****p < 0.0001$. Source data are provided as a Source data file.

TSC2-deficient cells, we downregulated TFEB by siRNA and observed decreased *GPNMB* promoter activity (Supplementary Fig. 4a), GPNMB protein expression (Supplementary Fig. 4b), as well as decreased expression of multiple lysosomal genes including *GPNMB*, ATPase H + transporting VO subunit D2 (*ATP6V0D2*), Interleukin 33 (*IL-33*) *NPC1* and Mucolipin1 (*MCOLN1*) (Supplementary Fig. 4c).

**Tfeb knockdown decreases proliferation of Tsc2-deficient cells.** To understand better the functions of TFEB in TSC, we knocked down *Tfeb* in *Tsc2*-expressing and deficient MEFs with two

different shRNAs (Fig. 4a). Downregulation of *Tfeb* decreased the proliferation of $Tsc2^{-/-}$ MEFs, with a 25% decrease for *shTfeb #1* and a 45% decrease for *shTfeb#2* at 72 h (Fig. 4b). *Tfeb* downregulation did not affect the growth of $Tsc2^{+/+}$ MEFs (Fig. 4b). Rapamycin (20 nM) had no additional effect on the proliferation of *Tsc2*-deficient cells with *Tfeb* downregulation (Supplementary Fig. 5a), although it decreased the expression of *Tfeb* by about 2-fold (Supplementary Fig. 5b). In vivo, the subcutaneous growth of *Tsc2*-null MEFs was decreased by 3-fold in cells with *Tfeb* downregulation (Fig. 4c). Taken together these data establish *Tfeb* as a driver of *Tsc2*-deficient cell proliferation in vitro and in vivo.

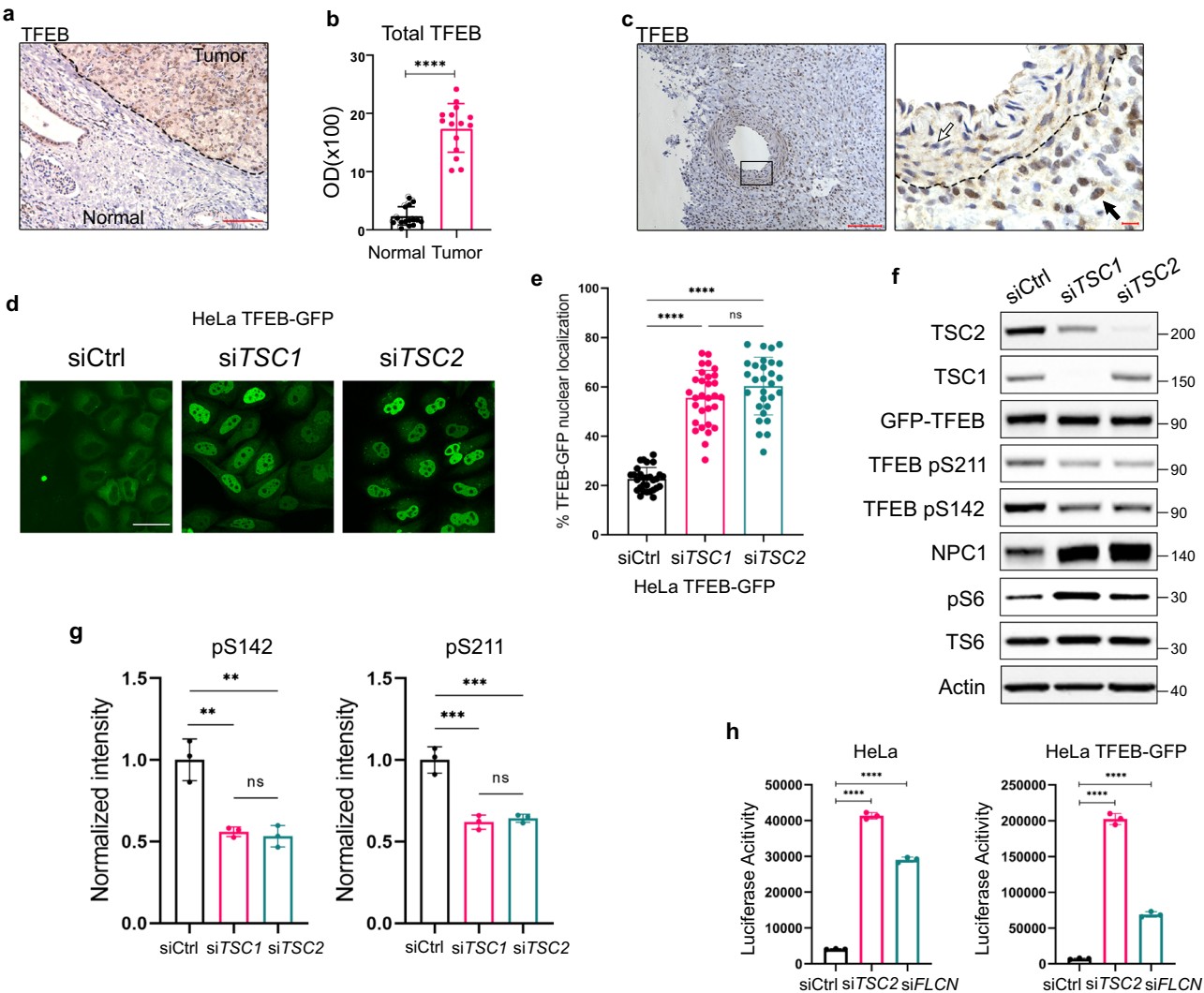

**Fig. 2 *TSC2* loss induces nuclear localization and increased transcriptional activity of TFEB. a** TFEB is increased and primarily nuclear in TSC-associated renal cell carcinoma. The dashed line shows the boundary between tumor and normal adjacent kidney. Scale bar = 100 μm. **b** TFEB optical density quantified from a, (15 measurements on 5 random areas of tumor and normal adjacent kidney quantified per section in 3 patient samples). **c** TFEB is primarily nuclear in human angiomyolipoma compared to blood vessel. The dashed line shows the boundary between tumor cells on the right and a blood vessel on the left. The black arrow indicates nuclear TFEB in a tumor cell. The white arrow indicates cytoplasmic TFEB in the cells of the blood vessel wall (*n* = 3 patient samples). Left image scale bar = 100 μm, right image scale bar = 10 μm. **d**, **e** HeLa-TFEB-GFP cells were transfected with *TSC1* or *TSC2* siRNA for 72 h and visualized with confocal live imaging. Scale bar = 50 μm (**d**). The nuclear/cytoplasmic ratio of GFP was quantified using ImageJ as described in methods, (*n* = 3 random fields per condition, 29 cells for Ctrl siRNA, 30 cells for *TSC1* siRNA and 29 cells for *TSC2* siRNA were analyzed) (**e**). **f** Representative immunoblotting of HeLa-TFEB-GFP cells transfected with *Ctrl, TSC1* or *TSC2* siRNA for 72 h (*n* = 3 biological replicates per condition). Blot was analyzed by staining with the indicated antibodies, phospho-S6 (S235/S236) is the indicator of increased mTORC1 activity, NPC1 is the indicator of TFEB transcriptional activity. **g** Densitometry for phospho TFEB S142 and phospho TFEB S211 was performed using ImageJ and normalized to total TFEB-GFP (*n* = 3 biological replicates per condition). **h** Luciferase activity of HeLa and HeLa-TFEB-GFP stably expressing the *GPNMB* luciferase reporter and transfected with *TSC2* or *FLCN* (as positive control) siRNAs for 72 h (*n* = 3 biological replicates per condition). Graphs are presented as mean ± SD. Statistical analyses were performed using two-tailed Students *t*-test or one-way ANOVA if more than two groups, **$p < 0.01$, ***$p < 0.001$, ****$p < 0.0001$. Source data are provided as a Source data file.

**FLCN and TSC2 coordinately regulate TFEB localization, activity, and phosphorylation.** FLCN is a GAP for RAGC/D, converting RAGC/D from the inactive (GTP)-bound to the active (GDP)-bound form, thereby facilitating the recruitment of TFEB to the surface of the lysosome where it can be phosphorylated by mTORC1[19–23,39]. Interestingly, we found that *FLCN* expression is increased ~2-fold upon *TSC2* downregulation in Hela and Hela-TFEB-GFP cells (Fig. 5a). To examine the relationship between TSC2 and FLCN, *TSC2* and *FLCN* were downregulated alone and in combination in HeLa, and HeLa TFEB-GFP cells using siRNA. The combined downregulation of *TSC2* and *FLCN* resulted in stronger nuclear localization of TFEB-GFP (Fig. 5b, c), a further increase in *GPNMB* activity (Fig. 5d), a further decrease in S211 phosphorylation (Fig. 5e, f), and a higher expression of lysosomal genes when compared to single gene knockdowns of either *TSC2* or *FLCN* (Fig. 5g). In parallel, we found that overexpression of FLCN (Myc-FLCN) partially restored TFEB phosphorylation at S142 and S211 in HEK293T cells after *TSC2* downregulation by

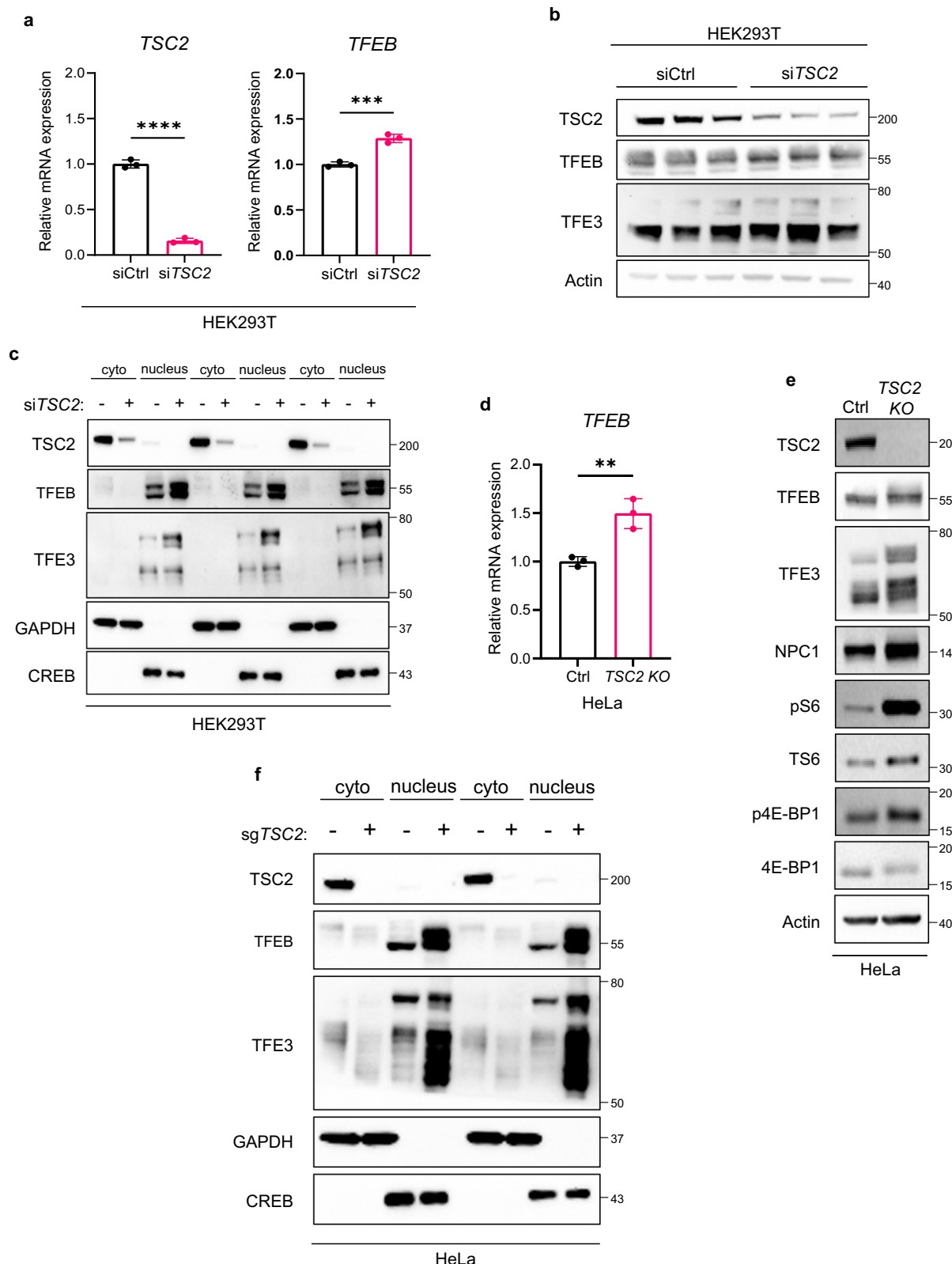

siRNA (Fig. 5h, i). Further work using CRISPR-mediated knockout of TSC2 and FLCN will be important to complement these siRNA-based findings.

**Activation of RAGC GTPase is necessary and sufficient to localize TFEB into the cytoplasm in *TSC2*-deficient cells.** Our data showing that knockdown of either *FLCN* or *TSC2* has similar

effects on TFEB phosphorylation, localization, and activity led us to focus on the RAG GTPases in *TSC2*-deficient cells. We found that *RAGC* and *RAGD* were increased at the mRNA level in HeLa cells after knockdown of either *TSC2* or *FLCN*, and even further increased with downregulation of both *TSC2* and *FLCN* (Supplementary Fig. 6a). In HeLa-TFEB-GFP (which do not express detectable level of *RAGD*), a similar pattern was found for the expression of *RAGC* at

**Fig. 3 Endogenous TFEB and TFE3 are enriched in the nucleus of TSC2-deficient HEK293T and HeLa cells. a** qRT-PCR analysis of *TSC2* and *TFEB* expression in HEK293T cells transfected with control or *TSC2* siRNA for 72 h ($n = 3$ biological replicates per condition), $p < 0.0001$ for *TSC2* and $p = 0.0008$ for *TFEB*. **b** Immunoblot analysis (biologic triplicates) of whole-cell lysates of HEK293T cells transfected as in (**a**) with indicated antibodies and used for fractionation in (**c**). **c** Immunoblot analysis (biological triplicates) of TFEB and TFE3 in cytoplasmic (cyto) and nuclear (nucleus) fractions of HEK293T cells transfected as in (**a**), GAPDH and CREB were used as markers of cytoplasmic and nuclear fraction, respectively. **d** qRT-PCR analysis of *TFEB* expression in HeLa cells with non-targeting control (*Ctrl*) or *TSC2* CRISPR knock-out (*TSC2 KO*) ($n = 3$ biological replicates), $p = 0.0062$. **e** Representative immunoblotting of TFEB and TFE3 in whole-cell lysates of *Ctrl* and *TSC2 KO* HeLa cells used for fractionation in (**f**), with phospho-S6 (S235/S236) and p4E-BP1 (Thr37/46) as indicators of mTORC1 activity and NPC1 as an indicator of TFEB transcriptional activity. **f** Immunoblotting of TFEB and TFE3 in cytoplasmic (cyto) and nuclear (nucleus) fractions of HeLa *Ctrl* and *TSC2 KO* cells, GAPDH and CREB were used as markers of cytoplasmic and nuclear fraction, respectively ($n = 3$ biological replicates per condition). Graphs are presented as mean ± SD. Statistical analyses were performed using two-tailed Students *t*-test, **$p < 0.01$, ***$p < 0.001$, ****$p < 0.0001$. Source data are provided as a Source data file.

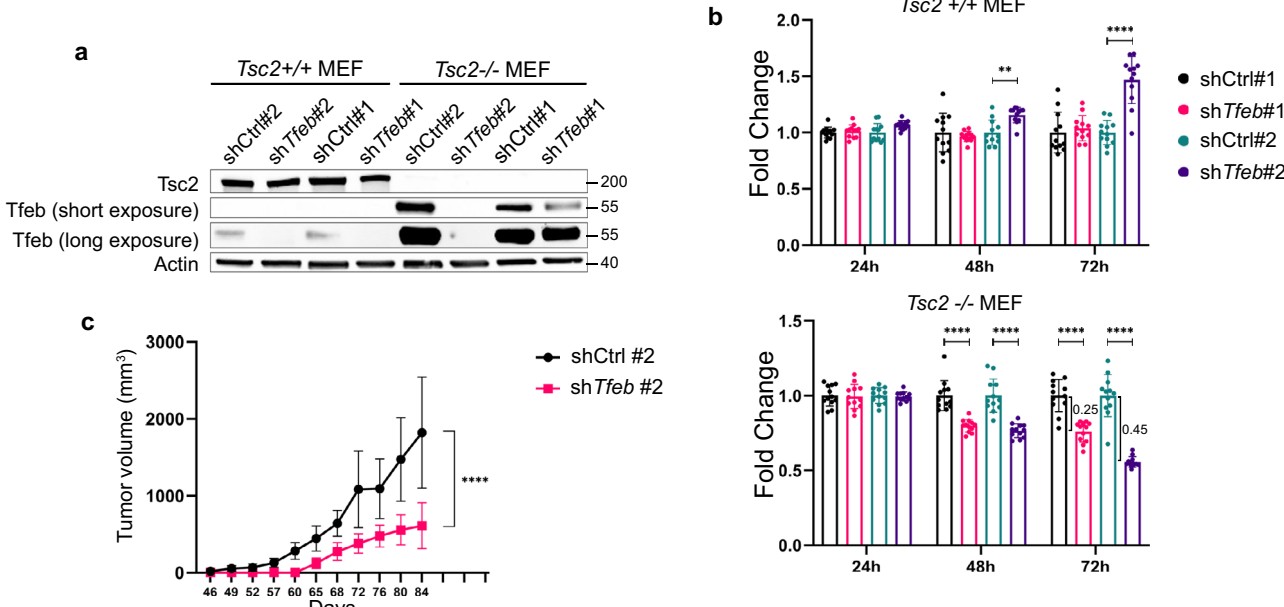

**Fig. 4 *Tfeb* downregulation decreases proliferation of *Tsc*-deficient cells. a** Confirmation of shRNA knockdown of TFEB in *Tsc2*[+/+] and *Tsc2*[−/−] MEFs ($n = 3$ biological replicates per condition). **b** Proliferation of *Tsc2*[+/+] and *Tsc2*[−/−] MEFs with *Ctrl* or *Tfeb* shRNA assessed by crystal violet staining ($n = 12$ biological replicates per condition). **c** Tumor volume of *Tsc2*[−/−] MEFs with *Ctrl* or *Tfeb* shRNA#2 subcutaneously injected into immunodeficient mice ($n = 10$ each group). Graphs are presented as mean ± SD. Statistical analyses were performed using two-tailed Students *t*-test, **$p < 0.01$, ****$p < 0.0001$. Source data are provided as a Source data file.

the mRNA and protein levels (Supplementary Fig. 6b, c). Interestingly, *RAGC/D* have been shown to be TFEB targets[40].

The increase in *RAGC/D* expression in *TSC2*-deficient cells, together with the increase in *FLCN* expression, suggest a potential role of the RAG GTPases in the nuclear localization of TFEB in TSC. To determine if the localization of TFEB in *TSC2*-deficient cells is RAG-dependent we overexpressed wild-type RAGA/C and constitutively active (CA) RAGA/C (RAGA[Q66L]-GTP bound, RAGC[S75N]-GDP bound)[41] in HeLa-TFEB-GFP cells with *TSC2*-downregulation. CA active RAGA/C, but not wild-type RAGA/C, decreased the nuclear localization of TFEB in HeLa-TFEB-GFP cells with *TSC2*-downregulation (Fig. 6a). To dissect which component of the RAG heterodimer was primarily responsible for the decreased nuclear localization of TFEB-GFP, we individually expressed wild-type (WT) RAG A, WT RAG C, CA RAG A, and CA RAG C. CA RAGC (but not WT RAGA, WT RAGC or CA RAGA) decreased nuclear TFEB-GFP levels (Fig. 6b, Supplementary Fig. 7a, b). CA RAGC increased the levels of TFEB phosphorylation at S211 in cells with *TSC2* downregulation (Fig. 6c) and CA RAGC alone was as efficient as the CA RAGA/C combination in relocalizing TFEB-GFP to the cytoplasm (Supplementary Fig. 8a, b) and in increasing TFEB phosphorylation at S142/211 (Supplementary Fig. 8c). Treatment

with Torin1, an mTOR kinase inhibitor (250 nM, 6 h), resulted in nuclear localization of TFEB in both siCtrl and siTSC2 HeLa-TFEB-GFP cells expressing CA RAGC (Supplementary Fig. 9). These data indicate that the localization of TFEB in *TSC2*-deficient cells is both RAGC and mTOR-dependent (Fig. 6d).

## Discussion

In this study we demonstrate that TFEB is predominantly nuclear in human TSC lesions and in *TSC*-deficient cells with both acute and chronic downregulation of *TSC2* and *TSC1*, where it drives lysosomal biogenesis and cell growth in vitro and in vivo. In *TSC*-deficient cells, TFEB is hypo-phosphorylated at the mTORC1-dependent sites. TFEB's nuclear localization and hypo-phosphorylation in TSC are unexpected because *TSC*-deficient cells have high mTORC1 activity, and phosphorylation of TFEB by mTORC1 is a well-established mechanism of TFEB's cytoplasmic sequestration[13,15–17]. Prior analyses of TFEB localization in *TSC2*-deficient cells have shown variable results, with some studies showing primarily nuclear localization[28,29] and others primarily cytoplasmic localization[15,24,27]. Of note, the prior studies focused on cultured cell models, while our work included also mouse and human tumor specimens of TSC, cellular models of acute and chronic loss of *TSC2*, and multiple methods of *TSC2*

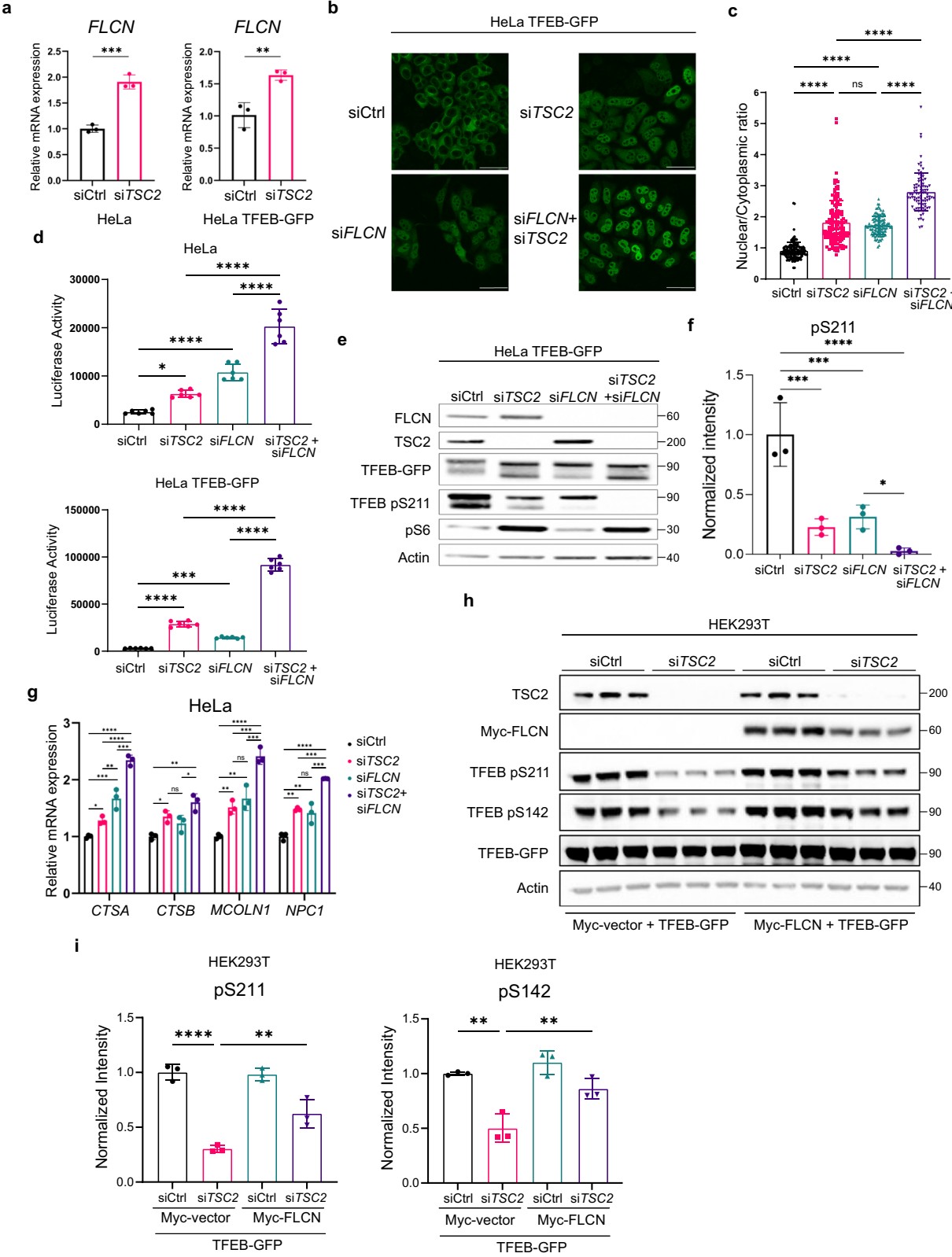

downregulation (littermate-derived $Tsc2^{+/+}$ and $Tsc2^{-/-}$ MEFs, siRNA, and CRISPR/Cas9 downregulation of *TSC2*). The reasons for the differing results are unclear at this time but could reflect differences in nutrient conditions and/or the duration and extent of *TSC2* downregulation. Taken together, our data indicate that a non-canonical regulatory mechanism is responsible for TFEB's nuclear localization in TSC.

Lysosomes are an emerging driver of tumorigenesis[34–36]. *TFEB* and *TFE3* are oncogenes, with translocations involving splicing genes and *TFE3* (or *TFEB*) causing a particularly aggressive RCC that disproportionally affects children and young adults[42,43]. These translocation RCC, which were previously referred to as "TSC-like"[42], may reflect key similarities to RCC in TSC patients[3,44], in which we have found high levels of nuclear TFEB

**Fig. 5 TSC2 and FLCN cooperate in the regulation of TFEB phosphorylation, nuclear translocation, and lysosomal gene expression. a** qRT-PCR analysis of *FLCN* in HeLa and HeLa-TFEB-GFP cells transfected with *Ctrl* or *TSC2* siRNA for 72 h (*n* = 3 biological replicates per condition), *p* = 0.0005 for HeLa and *p* = 0.007 for HeLa TFEB-GFP cells. **b, c** HeLa-TFEB-GFP cells were transfected with indicated siRNAs for 72 h and analyzed by confocal imaging after fixation and staining for GFP. Scale bar = 50 μm, **b**, nuclear/cytoplasmic ratio of TFEB-GFP as quantitated with Cell Profiler is shown in (**c**) (146 cells were analyzed in Ctrl siRNA, 140 cells in *TSC2* siRNA, 120 cells in *FLCN* siRNA and 88 cells in *TSC2+FLCN* siRNA were analyzed in *n* = 3 biological replicates). **d** Luciferase activity of HeLa and HeLa-TFEB-GFP stably expressing the *GPNMB* luciferase reporter and transfected with indicated siRNAs for 72 h (*n* = 6 biological replicates per condition). **e, f** Representative immunoblotting of phosphorylated TFEB at S211 in HeLa-GFP-TFEB cells after downregulation of *TSC2*, *FLCN*, or both analyzed by staining with the indicated antibodies, with phospho-S6 (S235/S236) as an indicator of mTORC1 activity (**e**), band intensity quantitated using ImageJ and normalized to total TFEB-GFP (*n* = 3 biological replicates per condition) (**f**). **g** Expression of lysosomal genes in HeLa cells after siRNA downregulation for 72 h of *TSC2*, *FLCN*, or both (*n* = 3 biological replicates each condition). **h, i** Overexpression of myc-FLCN in HEK293T cells with *TSC2* downregulation by siRNA increases TFEB-GFP phosphorylation at S211 and S142 (**h**), band intensity quantitated using ImageJ and normalized to total TFEB-GFP (**i**) (*n* = 3 biological replicates). Graphs are presented as mean ± SD. Statistical analyses were performed using two-tailed Students *t*-test, or one-way ANOVA if more than two groups, **p* < 0.05, ***p* < 0.01, ****p* < 0.001, *****p* < 0.0001. Source data are provided as a Source data file.

and high expression of the lysosomal protein NPC1. Moreover, TFEB has been shown to be a primary driver of pancreatic adenocarcinoma[45]. Our data support the concept that activation of TFEB is a key driver of renal tumorigenesis in TSC.

TFEB and its family members TFE3 and MITF may also be involved in other manifestations of TSC via enhanced lysosomal biogenesis and/or other mechanisms. Cathepsin K (a lysosomal enzyme) is known to be upregulated in pulmonary LAM[46] and multiple lysosomal genes are increased in TSC-associated subependymal giant cell astrocytomas compared with normal brain[47]. The hypothesis that TFEB and lysosomes are directly involved in the pathogenesis of TSC via a non-canonical RAGC-dependent mechanism challenges the concept that hyperactivity of mTORC1 to its canonical substrates is the unique driver of tumor formation in TSC.

Increased nuclear TFEB is a hallmark of BHD syndrome[24], which has some clinical similarity to TSC (both diseases are associated with chromophobe/oncocytic RCCs, benign facial skin tumors, and cystic lung disease)[48,49]. BHD is caused by mutations in *FLCN*, a known GAP for RAGC/D[20–22]. Levels of mTORC1 activity vary considerably between different in vitro and in vivo models of BHD, with some *FLCN*-deficient cells showing lower mTORC1 activity[50,51] and others showing higher mTORC1 activity[24,52,53]. As a GAP for RAGC/D, *FLCN* knockdown would be predicted to result in lower mTORC1 activity[51], on the other hand, FLCN loss has been shown to increase the expression of RAGD through TFEB and thereby boost RAG GTPase-dependent mTORC1 activation[40], in fact hyperactive mTORC1 has been observed in BHD-associated RCC[54]. We found increased expression of RAGC and RAGD in both *TSC2* and *FLCN*-deficient HeLa cells, with an even greater increase in RAGC/D expression in cells with downregulation of both *TSC2* and *FLCN*. The increase in RAGC/D expression in *TSC2*-deficient cells could represent an additional mechanism by which the absence of *TSC2* sustains high mTORC1 activity, similarly to what has been proposed for *FLCN* loss[40,55].

Our discovery that constitutively active RAGC, but not wild-type RAGC, is sufficient to induce cytoplasmic TFEB localization in *TSC2*-deficient cells further supports a model in which FLCN and TSC2 act in parallel to regulate TFEB via the activity of RAGC/D. It is possible that in a *TSC*-deficient setting, impaired RAGC/D activity is responsible for the inability of mTORC1 to phosphorylate TFEB, as has been shown for *FLCN*-deficient cells[20,24,56]. Moreover, the fact that combined loss of *FLCN* and *TSC2* induced even stronger nuclear localization of TFEB, while overexpression of FLCN in *TSC2*-deficient cells increased the phosphorylation of TFEB at the mTORC1 sites, support a pathogenic link between TSC and BHD. Further work will be needed to determine if *Tfeb* inactivation can alleviate renal disease in genetically engineered mouse models of TSC, as has been recently demonstrated for BHD-associated renal disease[24].

In summary, we identified a non-canonical RAGC-dependent pathway through which loss of the TSC complex drives TFEB into the nucleus and increases lysosomal gene expression and cell proliferation. TFEB represents a previously unrecognized pathogenic link between the clinical manifestations of TSC and BHD and may represent a therapeutic target for the treatment of both diseases.

## Methods

**Cell culture.** *Tsc1*[+/+], *Tsc1*[−/−], *Tsc2*[+/+], *Tsc2*[−/−] MEFs were provided by Dr. David Kwiatkowski at Brigham and Women's Hospital Boston, MA, US. HeLa cells and HEK293T cells were purchased from ATCC. Hela-TFEB-GFP cells were developed by Shawn Ferguson[16]. All cells were grown in Dulbecco's Modified Eagle Medium (DMEM, Gibco/Thermo Fisher Scientific, Waltham, MA, USA) supplemented with 10% fetal bovine serum with 1% penicillin/streptomycin, with the exception of the *Tsc2*[+/+] and *Tsc2*[−/−] MEFs in Fig.1h and i, which were grown in 0.1% FBS for 96 h. Rapamycin R-5000 (20 nM) and Torin1 T-7887 (250 nM) were purchased from LC Laboratories.

**Generation of TFEB knockdown cell lines.** *Tfeb* mouse shRNA Lentiviral Transduction Particles (Sigma-Aldrich, clone TRCN0000085548 and Dharmacon, clone V3SVMM00_13136092) were used to downregulate *Tfeb* in *Tsc2*[+/+] and *Tsc2*[−/−] MEFs. Cells were transduced with Lentiviral particles for 24 h with polybrene (8 μg/ml) and selected with puromycin at 5 μg/ml.

**Generation of HeLa TSC2 CRISPR knock-out line.** Non-targeting control gRNA and *TSC2* gRNA plasmids were designed using pRP_gRNA_Cas9 plasmid backbone from Vector Builder. The expression of small guide RNAs and mCherry-tagged codon-optimized Cas9 from Streptococcus pyogenes was driven by U6 and modified chicken beta-actin (CBh) promoters respectively. Both non-targeting control and *TSC2* CRISPR plasmids included two small guide RNAs (GTGTAGTTCGACCATTCGTG and GTTCAGGATCACGTTACCGC for non-targeting control; TCCTTGCGATGTACTCGTCG and GACCCGGTCGTTAC-TAGGCC for *TSC2*). Plasmids were transiently expressed in HeLa cells using Lipofectamine 3000 (Life Technologies/Thermo Fisher) for 48 h and single-cell sorted for mCherry on 96-well plates. Single Clones were isolated and TSC2 knock-out was verified by immunoblotting.

**siRNA transfection.** The following siRNA reagents were used: *TSC2* Silencer Select siRNA (ThermoFisher, assay ID s502596), *TSC1* Silencer Select siRNA (Thermofisher, assay ID s526384), *FLCN* Silencer Select siRNA (ThermoFisher, assay ID s47320), *TFEB* Silencer Select siRNA (ThermoFisher, assay ID s15496) and non-targeting control Silencer Select siRNA (ThermoFisher, 4390844). All siRNA transfections were performed using Lipofectamine RNAiMax (Thermo-Fisher, LMRNA015).

**Cell viability.** Cell viability was assessed using crystal violet staining. One thousand cells were seeded on clear 96-well plates. Cells treated with Dmso or Rapamycin (20 nM) were fixed after 24, 48, 72 h with 10% formalin for 10 min and stained with 500 mg/L crystal violet (Sigma) solution for 20 min. Crystal violet was then eluted with methanol for 5 min in a plate shaker. Absorbance was measured at 540 nm.

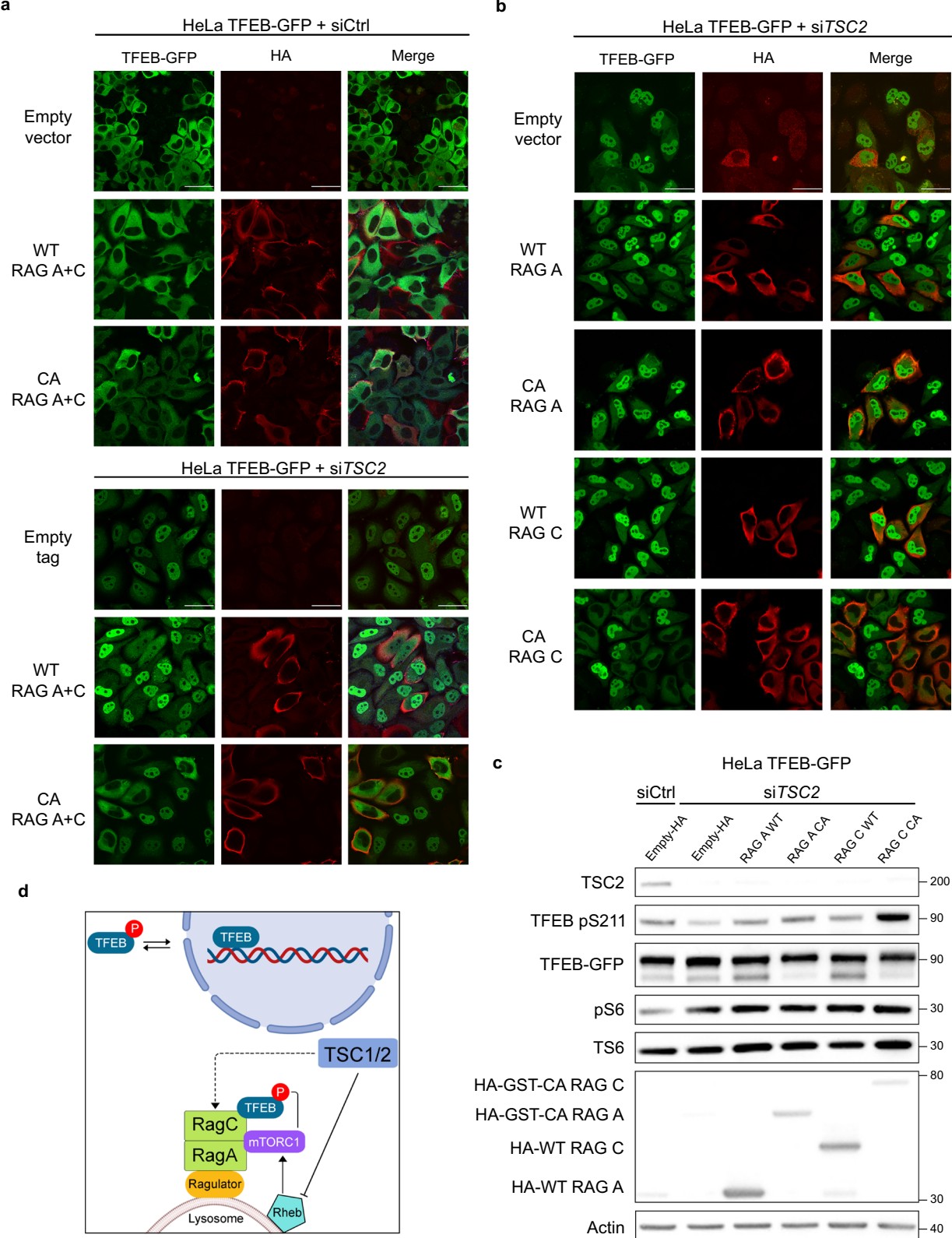

**Fig. 6 Activation of RAGC is sufficient to re-localize TFEB into the cytoplasm in *TSC2*-deficient cells. a** Immunofluorescent analysis of HeLa-TFEB-GFP cells after siRNA downregulation for 72 h transfected with wild-type (WT) RAGA plus WT RAGC vs. constitutively active (CA) RAGA (RAGA Q66L) plus CA RAGC (RAGC S75N) for 48 h (*n* = 3 biological replicates per condition). **b** Immunofluorescent analysis of HeLa-TFEB-GFP cells after *TSC2* siRNA downregulation as in (**a**), and individually transfected with WT RAGA, CA RAGA, WT RAGC, or CA RAGC for 48 h (*n* = 3 biological replicates per condition). **c** Representative immunoblot analysis of cells treated as in (**b**) with indicated antibodies (*n* = 3 biological replicates per condition). **d** Working model in which TSC2 regulates TFEB cytoplasmic/nuclear localization via RAGC (created with BioRender.com). Scale bars = 50 µm. Source data are provided as a Source data file.

**Animal studies**. Animal studies were approved by the Brigham and Women's Hospital Animal Care and Use Committee (IACUC) and conducted according to an approved protocol for the care and maintenance of laboratory animals. Mice were housed in an animal facility with 12 h light/12 h dark cycle at 72 °F and 40% humidity with unrestricted access to food and water. 18 months old female AJ $Tsc2$ +/− mice were used for electron microscopy of the kidneys. Xenografts tumors were generated by subcutaneously injecting $(2.0 \times 10^6)$ $Tsc2^{-/-}$ MEFs mixed with matrigel 1:1 (v/v) (Corning, 356237) to a final volume of 100–150 µl unilaterally into the shoulders of 8-week-old female NOD-scid IL2Rgamma$^{null}$ anesthetized mice using a 21 G needle. Mice were inspected weekly, and tumors were measured every three days by caliper once they became palpable and >100 m$^3$.

**Electron microscopy**. Kidney specimens from $Tsc2^{+/−}$ mice in the AJ genetic background at 18 months of age were fixed in 2% glutaraldehyde/2% paraformaldehyde in 0.1 M phosphate buffer for 3 h at 4 °C, postfixed in 1% osmium tetroxide in the same buffer solution, dehydrated in graded alcohols, and embedded in an Epon-Araldite mixture. Thin sections were stained with lead citrate and examined with a CM10 transmission electron microscope. Ten random images at the same magnification for each sample were utilized for lysosome quantitation in either cysts or normal adjacent kidney tissue.

**Protein extraction and western blot analysis**. Cells were washed with ice-cold PBS, scraped, and lysed on ice with 1× RIPA buffer. Lysates were normalized by concentration and each sample resolved on 4–12% Bis-Tris gel. Chemiluminescence was visualized with SuperSignal West Pico PLUS Chemiluminescent Substrate (Thermo Fisher Scientific, Waltham, MA, USA). To isolate the nuclear and the cytoplasmic protein fraction, CelLytic NuCLEAR Extraction Kit was used (Millipore Sigma, NACRES NA56). The following antibodies were used at 1:1000 dilution unless otherwise indicated: β-actin (Sigma-Aldrich, A5316), phosphorylated S6 ribosomal protein (pS6 Ser 235/236) (Cell Signaling Technology, 2211), total S6 ribosomal protein (TS6) (Cell Signaling Technology, 2217), phosphorylated 4E-BP1 (p4E-BP1 Thrn37/46) (Cell Signaling Technology, 2855), total 4E-BP1 (T4E-BP1) (Cell Signaling Technology, 9644), TSC2 (Cell Signaling Technology, 4308), TSC1 (Cell Signaling Technology, 4906), TFEB (1:3000) (Bethyl Laboratories, A303-673A), TFEB (Cell Signaling Technology, 32361), TFE3 (Cell Signaling Technology, 14779), Cathepsin K (CTS K) (Abcam, ab19027), NPC1 (Abcam, ab134113), FLCN (Cell Signaling Technology, 3697), GFP (1:5000) (Abcam, ab13970), phosphorylated TFEB S211 (Cell Signaling Technology, 37681), phosphorylated TFEB S142 (Millipore Sigma, ABE1971), RAGC (Cell Signaling Technology, 9480), MYC-Tag (Cell Signaling Technology, 2276), HA-tag (Cell Signaling Technology Inc., 3724S), GPNMB (Cell Signaling Technology, 38313), CREB (Cell Signaling Technology, 9197), GAPDH (Cell Signaling Technology, 5174).

**Plasmids**. The following plasmids were used in HeLa-TFEB-GFP cells: pRK5 HA-RAGA (#99710), pRK5-HA GST RAGA 66L (#19300), pRK5 HA- RAGC (#99718), and pRK5-HA GST RAGC 75L (#19305), were purchased from Addgene. The following plasmids were used in HeLa cells: TFEB-GFP wild type, TFEB-GFP S142A, TFEB-GFP S211A, TFEB-GFP S142A/S211A developed by Dr. Shawn Ferguson. The following plasmids were used in HEK293T cells: pEGFP-N1-TFEB (Addgene #38119), pCMV-Tag3B vector plasmid (Agilent) and pCMV-Tag3B expressing myc-FLCN plasmid[57]. All plasmids were expressed using Fugene6 reagent (Promega) for 48 h in immunofluorescence experiments and using Lipofectamine 3000 (Invitrogen) for 6 h in immunoblotting experiments.

**GPNMB reporter activity**. The GPNMB reporter was generated by PCR amplification of a 530 bp promoter fragment upstream of the GPNMB gene using genomic DNA from LNCaP cells as a template. The following forward and reverse primers, containing terminal NcoI sites and Kozak sequence, were used for amplification: GPNMB_promoter_F:CATGGccatggCCAACATAGTGAAACCTGCC; GPNMB_promoter_R:CATGGccatggtggcTGAATTCTCACGGACGCAGG. Following amplification, the fragment was cloned into a Gateway-compatible entry vector upstream of a NanoLuc luciferase cassette using a unique NcoI site. The GPNMB-NanoLuc reporter was then transferred into a promoterless Gateway-compatible lentiviral destination vector carrying Blasticidin resistance for use in downstream studies. Lentiviral particles were prepared using HEK293T cells as a packaging line. HeLa and Hela-GFP-TFEB cells were transduced with viral particles and selected using 5ug/ml Blasticidin S to create stable lines. To measure the GPNMB promoter activity cells were transfected with appropriate siRNA constructs for 48 h and then seeded on opaque-white 96-well plates, allowed to attach and luminescence was assessed using Nano-Glo Luciferase assay system (Promega) on Biotek Synergy HT multi-well plate reader. All data were normalized to cell seeding.

**mRNA extraction and real-time PCR**. mRNA was isolated using the RNeasy Plus Micro Kit with on-column genomic DNA-digest (Qiagen) according to the manufacturer's protocol. To generate cDNA from isolated and purified mRNA, Affinity Script quantitative PCR (qPCR) cDNA Synthesis Kit (Agilent Technologies) was used. Real-time PCR was conducted using StepOne Plus Realtime PCR Machine (Applied Biosystems) with TaqMan Real-Time PCR Master Mix (Thermo Fisher

Scientific). Gene expression was measured relative to β-actin and delta delta Ct (ΔΔCt) method was used to calculate the fold change differences of the experimental groups compared to the control group. TaqMan real-time PCR assays (Thermo Fisher Scientific) used for RT-PCR are listed in Supplementary Table 1.

**Live imaging**. Hela-TFEB-GFP cells were plated onto 35 mm glass bottom dishes (Mattek Life Sciences) and grown in the appropriate conditions, images were captured using Olympus Fluoview FV10i confocal microscope at 60x magnification in live cells.

**Immunofluorescence**. MEFs and HeLa-TFEB-GFP cells were plated onto 35 mm glass bottom dishes, fixed with 2% paraformaldehyde (PFA) for 15 min, permeabilized with 0.1% Triton X-100 for 5 min and washed three times with PBS. Cells were then blocked for 30 min in 1% BSA and incubated with primary antibodies, TFEB (1:200) (Cell Signaling Technology, 32361), TFE3 (1:200) (Millipore Sigma, HPA023881), GFP (1:1000) (abcam, ab13970), and HA-tag (1:1000) (Cell Signaling Technology, 3724S) in blocking buffer (1% BSA) for 1 h. Cells were then washed three times with PBS and stained with secondary antibody (1:1000 dilution) anti-Rabbit Alexa Fluor Red 568 (Life Technologies), in blocking buffer (1% BSA) and kept in the dark for 1 h. DAPI (4′, 6-diamidino-2-phenylindole) (Sigma-Aldrich) was used to visualize nuclei. Cells were washed again three times with PBS and mounted with VECTASHIELD Antifade Mounting Medium (Vector Laboratories). Confocal images in HeLa-TFEB-GFP cells were analyzed for percentage of nuclear localization in ImageJ, the background was removed from the channel corresponding to the protein of interest (TFEB-GFP) using the rolling ball radius method. The DAPI channel was used to generate nuclear mask of corresponding cells. The threshold of the DAPI image was adjusted, transformed to binary and then analyzed to produce nuclear regions of interest (ROI). This mask was then applied to the protein of interest (TFEB-GFP) channel and the total fluorescence intensity of the nuclear ROIs was measured. Phalloidin (1:1000) (Alexa Fluor 568 Phalloidin, A12380) channel was used to generate a mask of the whole cell of which an ROI was created and applied to the protein of interest channel to measure the total fluorescence intensity of the whole cell. The percentage of nuclear localization was calculated as described[58]. Three random images, each containing multiple cells, were used for quantification and statistical analysis in each condition.

Nuclear to cytoplasmic TFEB ratio in Fig. 5b, c was quantified using CellProfiler (https://cellprofiler.org/). DAPI staining was used to identify the nuclei and phalloidin staining (1:1000) was used to create a mask that defines the cytoplasmic compartment in each cell. The mean intensities of the nuclear and cytoplasmic regions were measured and used to calculate nuclear to cytoplasmic ratio.

**Immunohistochemistry**. Immunohistochemistry (IHC) was performed on formalin-fixed, paraffin-embedded, and sectioned kidneys from $Tsc2^{+/−}$ mice and human kidney and lung tissue samples. Human tissue samples were obtained with the approval of Partners Healthcare Human Research Committee. Briefly, slides were incubated at 65 C°, dewaxed with Xylene and ethanol and then rehydrated with H20. Antigen retrieval was performed in citrate buffer (ph = 6) by a heat-induced process using a Russell Hobbs pressure cooker, washed three times in H20, and then blocked with 5% goat serum in TBS for 1 h at room temperature. Sections were incubated overnight at 4 °C with primary antibodies: TFEB (1:150) (Cell Signaling Technology, 32361), NPC1 (1:400) (Abcam, ab134113), and HMB45 (1:100) (Dako, M0643) in blocking buffer. After washing three times in TBS, cells were incubated with ImmPRESS-HRP species-specific secondary antibodies (Vector Laboratories), washed again three times in TBS, and then incubated with ImmPACT DAB peroxidase substrate (Vector Laboratories, #SK4105). Finally, slides were counterstained with haematoxylin (Dako, #S3309), washed in tap water, dehydrated in ethanol and xylene prior to mounting with DPX (Sigma-Aldrich, #06522).

**Immunohistochemistry image analysis**. IHC slides were analyzed with ImageJ as described before[59]. In brief, the plug-in "color deconvolution" was executed on the digitalized IHC sections using the built-in vector HDAB, which separates the staining into 3 different panels with hematoxylin, DAB only image, and background. The DAB panel was used to measure the mean gray value of the region of interest. Then, the optical density (absorbance) was inferred by taking the $log_{10}$ of the ratio of the maximum value of an 8-bit image (255) over the measured mean gray value[60]. To measure nuclear DAB staining, we first used the hematoxylin panel and adjusted the threshold in order to generate a nuclear mask. This mask was then analyzed to generate nuclear ROIs which were applied to the DAB panel and the optical density was calculated.

**Statistical analyses**. All quantitative, normally distributed data for in vitro studies were analyzed for statistical significance using a Student's unpaired $t$-test, One-way ANOVA, and Tukey's post hoc tests when comparing more than two groups relative to a single factor, or two-way ANOVA and Tukey's post hoc tests when comparing more than two groups relative to more than one factor.

GraphPad Prism Software (GraphPad Prism version 8.3.1 for Windows; GraphPad Software, www.graphpad.com) was used.

# ARTICLE

**Reporting summary**. Further information on research design is available in the Nature Research Reporting Summary linked to this article.

## Data availability

All data are available from the corresponding author upon request and all unique materials generated (such as HeLa *TSC2* CRISPR KO line) are readily available from the authors. Source data for Figs. 1b, f, g, h, i, 2b, e, g, h, f, 3a–f, 4a–c, 5a, c–i, 6c and Suppl Figs. 1b, e, 4a–c, 5a, b, 6a–c, 7b, 8b, c are provided with this paper in a Source data file and a separate supplementary pdf file with uncropped western blots. Source data are provided with this paper.

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

## Acknowledgements
This work was supported by the Lucy Engles Fund for TSC and LAM Research (to E.P.H.), the Claudia Adams Program for Innovative Cancer Research (to S.R.V.), and a FAR grant from the Universita' Politecnica Delle Marche (to M.M.). We thank Craig A. Strahdee for providing parental luciferase reporter plasmids.

## Author contributions
N.A. and E.P.H. conceived the study. N.A., E.W.A., D.K., S.M.F., W.S. and E.P.H. designed the experiments. N.A., E.W.A., D.K. and A.S.N. performed all the in vitro experiments. E.R.G. and S.R.V. designed the GPNMB promoter. N.A. and H.F performed the in vivo experiments. H.C.L. harvested and prepared tissue for IHC and TEM. N.A., E.W.A. and H.J.L. performed and analyzed IHC experiments. M.M. performed and analyzed TEM experiments. N.A., E.W.A., D.K. and E.P.H. wrote the manuscript. E.P.H. supervised the study.

## Competing interests
The authors declare no competing interests.
