## [Peer Review File · Nature Communications]

REVIEWER COMMENTS

Reviewer #1 (Remarks to the Author):

The manuscript by the Henske group investigates the control of TFEB function in TSC mutant cells. This is a relevant topic as recent work by the Ballabio group has published in Nature the key role of TFEB in kidney cancer and malformations of folliculin mutants. This present study elaborates on the complexity of TFEB regulation, nicely complementing previous work or raising controversial issues. The authors propose that TFEB is also activated in TSC mutants and may participate to the kidney lesions in this genetic disease. Overall, the experiments are well performed and the data convincing. The conclusion that this is an evidence of mTOR-independent outcomes in TSC is not really demonstrated and should be smoothed (see below).

1- The claim that TFEB acts as a "TSC target that drives proliferation independently of mTORC1" should be demonstrated with rapamycin treatment or raptor deletion. Is tumor volume (Fig. 2k) and cell proliferation (Fig. 2j) sensitive to mTOR inhibition?

2- The authors show a massive induction of TFEB mRNA and protein levels in TSC mutant cells (extended Fig. 1a and Fig. 2i) without further comments. Is this induction mTOR dependent? Might the TFEB overexpression rather than a mild hypophosphorylation explain TFEB activation?

3- The staining of the lysosomal marker NPC1 appears nuclear in Fig. 1D and extended 1C? Is this expected? Any control of antibody specificity?

Reviewer #2 (Remarks to the Author):

In this manuscript, Alesi et al showed that loss of function of TSC2 is associated with enhanced lysosomal biogenesis due to TFEB activation. They suggest that this pathway may have a role in TSC pathogenesis, similar to recent findings regarding the loss of function of FLCN. The observation that mutations in TSC2 result in TFEB constitutive activation is interesting and suggests a novel mechanism underlying Tuberous Sclerosis. However, the interpretation of some of the data is not convincing and is not in agreement with the current literature on the topic. In addition, some of the data are not strong enough to support the authors' claims.

In particular, I have the following major concerns:

1. "Contrasting results" on TFEB subcellular localization in TSC2 KO cells.

The authors show that TFEB is constitutively localized to nuclei and active in TSC2-KO cells. As the authors mentioned in the introduction, previous analyses of TFEB activity in TSC2-deficient cells have led to contrasting results. Contrary to the current manuscript, TFEB localization was recently found to be unaffected upon TSC2 silencing (PMID: 32612235), as well as in TSC2-/- cells (PMID: 22343943), and in TSC2 KO neuronal progenitor cells (PMID: 28637240). When describing these "contrasting results" the authors should at least comment on possible explanations for these discrepancies (e.g. different siRNAs used, different KO cell lines, different experimental conditions, etc.). Related to this point, all experiments of TSC2 silencing were performed in a cell line that overexpresses TFEB-GFP. It would be important to repeat these experiments in another cell line and look at the endogenous TFEB. Additionally, Extended data Fig. 1A shows a striking increase (over 15-fold) in the levels of TFEB mRNA in TSC2 KO cells compared to controls. What is the mechanism underlying such an increase? Is the mechanism by which TSC2 loss of function causes TFEB activation mediated by TFEB nuclear translocation or is it mediated by TFEB transcriptional induction? This was not even discussed in the manuscript.

2. mTORC1 independence: upstream of TFEB

The authors propose that TFEB nuclear translocation in TSC is "mTORC1-independent". In particular,

the authors state that: "a non-canonical mTORC1-independent mechanism is responsible for TFEB's nuclear localization in TSC" and that "TFEB is nuclear in TSC2-deficient cells via a non-canonical, RagC activity-dependent mechanism". This "mTORC1 independence" claim is based on the observation that mTORC1 is hyperactive in TSC and, therefore, in this condition, TFEB should be retained in the cytoplasm. However, a recent paper clearly showed that FLCN loss of function, responsible for Birt-Hogg-Dube' syndrome, leads to impaired mTORC1 activity towards TFEB, whereas mTORC1 is hyperactive towards other substrates (PMID: 32612235). In my opinion, it is likely that the same situation also applies to TSC. Consistent with this possibility, the authors show that TFEB localization in TSC2-deficient cells is rescued upon the expression of constitutively active RagC. It is well established that the mechanism by which active Rag GTPases promote TFEB cytosolic localization is through the activation of mTORC1. Accordingly, Torin treatment in cells expressing active Rag GTPases completely impairs Rag-mediated cytosolic re-localization of TFEB (PMID: 32662822, 32989250). Thus, to claim that a "mTORC1-independent" mechanism modulates TFEB in their cellular system, the authors should test the effect of Torin, or of any other catalytic inhibitors of mTOR (such as AZD8055, PP242, etc.), in TSC2-deficient cells expressing active RagC. In the absence of such evidence, the authors cannot conclude that the mechanism is mTORC1-independent. If the authors are able to obtain convincing evidence for mTORC1 independence, they should then determine the mechanism responsible for TFEB nuclear translocation in TSC.

3. mTORC1 independence: downstream of TFEB

The authors state that "TFEB and lysosomes are directly involved in the pathogenesis of TSC via a non-canonical mTORC1-independent mechanism...". They also state that their study "challenges the concept that hyperactive mTORC1 is the primary driver of tumor formation in TSC". However, while data in Figure 2j-k do suggest that TFEB constitutive activation contributes to hyperproliferation in TSC2-/- cells, none of the experiments performed in this manuscript support that this TFEB-induced hyperproliferation is independent of mTORC1 hyperactivation. As a matter of fact, one of the key mechanisms by which TFEB has been shown to promote tumorigenesis is through transcriptional induction of RagC/D and consequent enhancement of mTORC1 activity (PMID: 28619945). Accordingly, TFEB depletion in a mouse model of BHD was recently shown to normalize mTORC1 activity and rescue the disease phenotype (PMID: 32612235). Thus, although additional TFEB-induced mechanisms may contribute to the TSC phenotype, mTORC1 hyperactivation is likely to be a key factor, downstream of TFEB, driving disease pathogenesis. Therefore, in the absence of any convincing evidence that TSC phenotype is independent of mTORC1 hyperactivity, the statements about mTORC1-independence throughout the manuscript should be removed.

4. Compliance with current literature: the role of FLCN.

Some of the concerns about mTORC1 independence in points 1 and 2 may have arisen due to misinterpretation of previously published findings. The author's state (lines 175-178, page 6): "Given that TSC2 and FLCN have opposing effects on mTORC1 activity, these data suggest that TSC2 and FLCN are both upstream of TFEB's regulation via a mechanism that does not directly involve mTORC1". They also state (lines 232-235, page 9): "FLCN mutations, therefore, result in lower mTORC1 activity, while tumors in TSC have higher mTORC1 activity. Our work suggests that hyperactivation of TFEB drives tumorigenesis in both diseases and may account for the similar phenotypes". The statement that FLCN mutations result in lower mTORC1 activity is inaccurate. mTORC1 hyperactivation is a well-known hallmark of BHD syndrome (PMID: 18182616, 18974783), a disorder caused by loss-of-function mutations of FLCN. Thus, loss of FLCN and of TSC2 are BOTH characterized by mTORC1 hyperactivation. Furthermore, recent studies revealed that the function of FLCN is crucial for mTORC1-mediated phosphorylation of TFEB, whereas it is largely dispensable for the phosphorylation of other mTORC1 substrates such as S6K (PMID: 27913603, 31672913, 32612235). Supporting this, the authors themselves did not find any differences in S6 phosphorylation upon depletion of FLCN (Fig3d).

It is very surprising that in the interpretation of their results the authors did not discuss or cite any of the above-mentioned studies, which are in line with the authors' own data. This should be corrected and those references should be properly discussed and cited in the paper.

5. Compliance with current literature: transcriptional induction of RagC/D.

The authors state (lines 244-248, page 93): "Since the increased lysosomal surface area is linked to increased mTORC1 activity, it is possible that TFEB-dependent lysosomal biogenesis is an additional driver of the hyperactive mTORC1 in TSC1/2 deficient cells". As stated above, previous studies have shown that TFEB promotes mTORC1 hyperactivation via transcriptional induction of RagC/D (PMID: 28619945, 30843872). Once again, it is very surprising that the authors did not discuss these studies and did not even consider this mechanism as a possible explanation for the TFEB-mediated mTORC1 hyperactivation observed in TSC. Also, there is not convincing experimental evidence supporting that the increased lysosomal surface is linked to mTORC1 activation.

6. Putative FLCN and TSC2 cooperation.

The data supporting cooperation between FLCN and TSC2 are very weak. The data in Figure 2b do not show a clear increase in TFEB nuclear localization in TSC2/FLCN-depleted cells compared to cells depleted of FLCN alone. In addition, the amount of nuclear TFEB observed in TSC2-depleted cells in Figure 3b is very different from that observed in the same cells treated in the same way in Figure 2d. Thus, the quantification shown in Figure 3c performed with 3 images only (as stated in the methods) is not sufficient to draw any conclusions. In addition to this, the amount of TFEB nuclear translocation in each treatment is likely dependent on the knockdown efficiency of every single protein. Moreover, the authors claim that overexpression of FLCN "rescues" TFEB phosphorylation in TSC2-silenced cells but the data shown are not convincing since total levels of TFEB-GFP are higher in TSC2-silenced cells upon FLCN overexpression relative to TSC2-silenced cells transfected with myc-vector in Fig 3g. Therefore, claiming that the two proteins cooperate in the modulation of TFEB without the use of double-ko cells and, above all, without providing a clear mechanism is merely speculative and is not supported by solid data. The authors should provide solid data and a clear mechanism showing how FLCN and TSC2 co-ordinately modulate TFEB activity. Alternatively, this part should be removed from the manuscript.

7. TFEB as a driver of TSC phenotype.

The data obtained by silencing TFEB in allografts generated with TSC2^{-/-} MEFs (in Fig 2k) suggest that TFEB is an important mediator of TSC phenotype. However, to obtain conclusive evidence the authors should generate tissue-specific TSC2/TFEB double-ko mice. The authors should at least discuss this approach.

Reviewer #3 (Remarks to the Author):

The paper by Alesi et al describe a novel role for the Tuberous Sclerosis Complex in the regulation of lysosomal biogenesis. The authors report that, in TSC2-null mice and cells, a striking upregulation of lysosomal gene expression correlates with constitutive nuclear translocation of the master regulator of lysosomal biogenesis, TFEB. The authors further show that TFEB dysregulation upon TSC loss resembles that caused by loss of the FLCN protein, a GTPase Activating Protein (GAP) for the RagC GTPase. Accordingly, forced expression of a GTP-locked RagC mutant rescues the lysosomal upregulation triggered by TSC loss. Based on these data and the similarities with FLCN loss, the authors conclude that constitutive upregulation of lysosomal biogenesis through TFEB dysregulation could be a novel driver mechanism in Tuberous Sclerosis.

The manuscript is potentially interesting as it reveals a novel facet of the relationship between the mTORC1 pathway and lysosomal biogenesis, with potential relevance to tumorigenesis. However, the manuscript suffers from conceptual weaknesses and lack of mechanism in some key points. In particular, the claim that regulation of TFEB by TSC is mTORC1-independent is unsupported mechanistically and is logically inconsistent with the authors' own experiments.

Specific comments:

1- Given the largely overlapping regulatory mechanisms between TFEB and TFE3 (i.e. PMID: 24448649), it is unclear why only TFEB and not TFE3 is constitutively nuclear in TSC KO cells.

2- Although upregulation of TFEB target genes in TSC cells correlates with increased TFEB localization in the nucleus, the two observations should be causally linked by showing that TFEB knockdown in TSC cells reverses the upregulation of lysosomal genes, both by qPCR and using the GPMNB luciferase assay.

3- By contrasting the coherent effects of TSC2 and FLCN deletion toward TFEB with their opposing roles in general mTORC1 signaling, the authors conclude that FLCN and TSC2 act on TFEB via mechanisms that do not involve mTORC1. This statement is problematic because it is not backed up by any mechanistic evidence. For example, does a phosphor-null TFEB specifically in the mTORC1 sites (S211A, S142A) fail to correct the upregulation of lysosomal target genes caused by TSC or FLCN loss?

4- Related to the previous point, if expressing RagC-CA in TSC cells increases TFEB phosphorylation and inhibits TFEB target genes, isn't this evidence that regulation of TFEB by TSC is in fact mTORC1-dependent? Or are the authors proposing that S211 is phosphorylated by another protein kinase and, if so, which one?

We are grateful to the Reviewers for their comments and suggestions, which we believe have strengthened our manuscript. We have performed additional experiments to confirm and validate our findings. The revised manuscript contains 22 new figure panels (Figure 3b, c, d, h, i, Supplementary Fig. 1a, 1d, 2b, 2c, 3a-f, 4, 5a-c, 6a, 6b, 10). We have also incorporated additional references and comments suggested by the reviewers in order to clarify the findings and provide a more complete interpretation of the data.

Our point-by-point responses are detailed below.

Reviewer #1 (Remarks to the Author):

The manuscript by the Henske group investigates the control of TFEB function in TSC mutant cells. This is a relevant topic as recent work by the Ballabio group has published in Nature the key role of TFEB in kidney cancer and malformations of folliculin mutants. This present study elaborates on the complexity of TFEB regulation, nicely complementing previous work or raising controversial issues. The authors propose that TFEB is also activated in TSC mutants and may participate to the kidney lesions in this genetic disease. Overall, the experiments are well performed and the data convincing. The conclusion that this is an evidence of mTOR-independent outcomes in TSC is not really demonstrated and should be smoothed (see below).

1. *The claim that TFEB acts as a “TSC target that drives proliferation independently of mTORC1” should be demonstrated with rapamycin treatment or raptor deletion. Is tumor volume (Fig. 2k) and cell proliferation (Fig. 2j) sensitive to mTOR inhibition?*

Response: In new experiments, we found that *TFEB* knockdown by shRNA does not further decrease the proliferation of Rapamycin-treated *TSC2*-deficient cells (Supplementary Fig. 6a). We have also removed the phrase “independently of mTORC1” and replaced this with “via a non-canonical mechanism.”

2. *The authors show a massive induction of TFEB mRNA and protein levels in TSC mutant cells (Supplementary Fig. 1a and Fig. 2i) without further comments. Is this induction mTOR dependent? Might the TFEB overexpression rather than a mild hypophosphorylation explain TFEB activation?*

Response: In new experiments we show that Rapamycin and Torin1 treatment decrease *Tfeb* expression by about 50% (Supplementary Figure 6b).

High levels of TFEB are seen in *Tsc2*-null MEFs, but not in other cellular models of *TSC1/2*-deficiency, including *Tsc1*-null MEFs (Supplementary Fig. 1b), and HeLa and HEK293T cells in which *TSC2* is downregulated (Supplementary Fig. 3a, b, d, e). In all models, including those in which TFEB is mildly upregulated, TFEB is primarily nuclear in the absence of *TSC2*. Therefore, elevated expression does not appear to be the mechanism of TFEB activation in TSC.

3. *The staining of the lysosomal marker NPC1 appears nuclear in Fig. 1D and extended 1C? Is this expected? Any control of antibody specificity?*

Response: Higher magnification images (Supplementary Fig. 1a, d) show more clearly that NPC1 staining is perinuclear.

Reviewer #2 (Remarks to the Author):

In this manuscript, Alesi et al showed that loss of function of TSC2 is associated with enhanced lysosomal biogenesis due to TFEB activation. They suggest that this pathway may have a role in TSC pathogenesis, similar to recent findings regarding the loss of function of FLCN. The observation that mutations in TSC2 result in TFEB constitutive activation is interesting and suggests a novel mechanism underlying Tuberous Sclerosis. However, the interpretation of some of the data is not convincing and is not in agreement with the current literature on the topic. In addition, some of the data are not strong enough to support the authors' claims. In particular, I have the following major concerns:

1. "Contrasting results" on TFEB subcellular localization in TSC2 KO cells.

The authors show that TFEB is constitutively localized to nuclei and active in TSC2-KO cells. As the authors mentioned in the introduction, previous analyses of TFEB activity in TSC2-deficient cells have led to contrasting results. Contrary to the current manuscript, TFEB localization was recently found to be unaffected upon TSC2 silencing (PMID: 32612235), as well as in TSC2^{-/-} cells (PMID: 22343943), and in TSC2 KO neuronal progenitor cells (PMID: 28637240). When describing these "contrasting results" the authors should at least comment on possible explanations for these discrepancies (e.g. different siRNAs used, different KO cell lines, different experimental conditions, etc.). Related to this point, all experiments of TSC2 silencing were performed in a cell line that overexpresses TFEB-GFP. It would be important to repeat these experiments in another cell line and look at the endogenous TFEB. Additionally, Extended data Fig. 1A shows a striking increase (over 15-fold) in the levels of TFEB mRNA in TSC2 KO cells compared to controls. What is the mechanism underlying such an increase? Is the mechanism by which TSC2 loss of function causes TFEB activation mediated by TFEB nuclear translocation or is it mediated by TFEB transcriptional induction? This was not even discussed in the manuscript.

Response: As suggested, we have included a discussion of why prior work may have shown contrasting results and added the three citations that were not previously referenced:

PMID:32612235 A substrate-specific mTORC1 pathway underlies Birt-Hogg-Dubé syndrome. Napolitano, Ballabio, et al. Nature, 2020.

PMID: 22343943, A lysosome-to-nucleus signaling mechanism senses and regulates the lysosome via mTOR and TFEB. Settembre, Ballabio, et al. EMBO J., 2012.

PMID: 28637240, TFEB activation restores migration ability to Tsc1-deficient adult neural stem/progenitor cells. Magini, Emiliani, et. Al. Hum Mol Genet., 2017.

The new text is below:

"Prior analyses of TFEB localization in TSC2-deficient cells have shown variable results, with some studies showing primarily nuclear localization and others primarily cytoplasmic localization. Of note, the prior studies focused on cultured cell models, while our work included also mouse and human tumor specimens of TSC, cellular models of acute and chronic loss of TSC2, and multiple methods of TSC2 downregulation (littermate-derived Tsc2^{+/+} and Tsc2^{-/-} MEFs, siRNA, and CRISPR/Cas9 downregulation of TSC2). The reasons for the differing results are unclear at this time but could reflect differences in nutrient conditions and/or the duration and extent of TSC2 downregulation. Taken together, our data indicate that a non-canonical regulatory mechanism is responsible for TFEB's nuclear localization in TSC."

Regarding repeating the experiments in cells with endogenous TFEB, we now demonstrate that TSC2 downregulation induces the nuclear localization of TFEB in HeLa cells

expressing endogenous TFEB and in HEK293T cells expressing endogenous TFEB (Supplementary Fig. 3c, f). We also show that *TSC2* downregulation in HeLa cells with endogenous TFEB increase lysosomal gene expression and *GPNMB* promoter activity (Fig. 3d, g, 2h).

Regarding the mechanisms of *TFEB* mRNA regulation, in new experiments (Supplementary Figure 6b) we show that Rapamycin and Torin1 treatment decrease *Tfeb* expression by about 50%. High levels of *TFEB* expression are seen in *Tsc2*-null MEFs, but not in other cellular models of *TSC1/2*-deficiency, including *Tsc1*-null MEFs (Supplementary Fig. 1b), and HeLa and HEK293T cells in which *TSC2* is downregulated (Supplementary Fig. 3a, b, d, e). In all models, including those in which *TFEB* mRNA is mildly upregulated, TFEB is primarily nuclear in the absence of *TSC2*. Therefore, elevated expression does not appear to be the mechanism of TFEB activation in TSC.

2. mTORC1 independence: upstream of TFEB

The authors propose that TFEB nuclear translocation in TSC is “mTORC1-independent”. In particular, the authors state that: “a non-canonical mTORC1-independent mechanism is responsible for TFEB’s nuclear localization in TSC” and that “TFEB is nuclear in TSC2-deficient cells via a non-canonical, RagC activity-dependent mechanism”. This “mTORC1 independence” claim is based on the observation that mTORC1 is hyperactive in TSC and, therefore, in this condition, TFEB should be retained in the cytoplasm. However, a recent paper clearly showed that FLCN loss of function, responsible for Birt-Hogg-Dube’ syndrome, leads to impaired mTORC1 activity towards TFEB, whereas mTORC1 is hyperactive towards other substrates (PMID: 32612235). In my opinion, it is likely that the same situation also applies to TSC. Consistent with this possibility, the authors show that TFEB localization in TSC2-deficient cells is rescued upon the expression of constitutively active RagC. It is well established that the mechanism by which active Rag GTPases promote TFEB cytosolic localization is through the activation of mTORC1. Accordingly, Torin treatment in cells expressing active Rag GTPases completely impairs Rag-mediated cytosolic re-localization of TFEB (PMID: 32662822, 32989250). Thus, to claim that a “mTORC1-independent” mechanism modulates TFEB in their cellular system, the authors should test the effect of Torin, or of any other catalytic inhibitors of mTOR (such as AZD8055, PP242, etc.), in TSC2-deficient cells expressing active RagC. In the absence of such evidence, the authors cannot conclude that the mechanism is mTORC1-independent. If the authors are able to obtain convincing evidence for mTORC1 independence, they should then determine the mechanism responsible for TFEB nuclear translocation in TSC.

Response: In new experiments, as suggested, we downregulated *TSC2* in HeLa-TFEB-GFP cells, transfected them with CA RAGC, and treated with Torin1. Under these conditions, TFEB is predominantly nuclear (Supplementary Fig. 10).

3. mTORC1 independence: downstream of TFEB

The authors state that “TFEB and lysosomes are directly involved in the pathogenesis of TSC via a non-canonical mTORC1-independent mechanism...”. They also state that their study “challenges the concept that hyperactive mTORC1 is the primary driver of tumor formation in TSC”. However, while data in Figure 2j-k do suggest that TFEB constitutive activation contributes to hyperproliferation in TSC2-/- cells, none of the experiments performed in this manuscript support that this TFEB-induced hyperproliferation is independent of mTORC1 hyperactivation. As a matter of fact, one of the key mechanisms by which TFEB has been shown to promote tumorigenesis is through transcriptional induction of RagC/D and consequent enhancement of mTORC1 activity (PMID: 28619945). Accordingly, TFEB depletion in a mouse model of BHD was recently shown to normalize mTORC1 activity and rescue the disease

phenotype (PMID: 32612235). Thus, although additional TFEB-induced mechanisms may contribute to the TSC phenotype, mTORC1 hyperactivation is likely to be a key factor, downstream of TFEB, driving disease pathogenesis. Therefore, in the absence of any convincing evidence that TSC phenotype is independent of mTORC1 hyperactivity, the statements about mTORC1-independence throughout the manuscript should be removed.

Response: The term “mTORC1 independent” has been replaced by “non-canonical” throughout the manuscript.

4. Compliance with current literature: the role of FLCN.

Some of the concerns about mTORC1 independence in points 1 and 2 may have arisen due to misinterpretation of previously published findings. The authors state (lines 175-178, page 6): “Given that TSC2 and FLCN have opposing effects on mTORC1 activity, these data suggest that TSC2 and FLCN are both upstream of TFEB’s regulation via a mechanism that does not directly involve mTORC1”. They also state (lines 232-235, page 9): “FLCN mutations, therefore, result in lower mTORC1 activity, while tumors in TSC have higher mTORC1 activity. Our work suggests that hyperactivation of TFEB drives tumorigenesis in both diseases and may account for the similar phenotypes”. The statement that FLCN mutations result in lower mTORC1 activity is inaccurate. mTORC1 hyperactivation is a well-known hallmark of BHD syndrome (PMID: 18182616, 18974783), a disorder caused by loss-of-function mutations of FLCN. Thus, loss of FLCN and of TSC2 are BOTH characterized by mTORC1 hyperactivation. Furthermore, recent studies revealed that the function of FLCN is crucial for mTORC1-mediated phosphorylation of TFEB, whereas it is largely dispensable for the phosphorylation of other mTORC1 substrates such as S6K (PMID: 27913603, 31672913, 32612235). Supporting this, the authors themselves did not find any differences in S6 phosphorylation upon depletion of FLCN (Fig3d). It is very surprising that in the interpretation of their results the authors did not discuss or cite any of the above-mentioned studies, which are in line with the authors’ own data. This should be corrected, and those references should be properly discussed and cited in the paper.

Response: We added and discussed the following references, as suggested:

PMID: 18182616, Kidney-targeted Birt-Hogg-Dube gene inactivation in a mouse model: Erk1/2 and Akt-mTOR activation, cell hyperproliferation, and polycystic kidneys. Baba, Schmidt, et al., J Natl Cancer Inst, 2008.

PMID:18974783, Deficiency of FLCN in mouse kidney led to development of polycystic kidneys and renal neoplasia. Chen, Teh, et al. Plos One, 2008.

PMID: 27913603, The tumor suppressor FLCN mediates an alternate mTOR pathway to regulate browning of adipose tissue. Wada, Arany, et al. Genes Dev, 2016.

PMID:31672913, Structural mechanism of a Rag GTPase activation checkpoint by the lysosomal folliculin complex. Lawrence, Zoncu, et al. Science, 2019.

PMID:32612235 A substrate-specific mTORC1 pathway underlies Birt-Hogg-Dubé syndrome. Napolitano, Ballabio, et al. Nature, 2020.

5. Compliance with current literature: transcriptional induction of RagC/D.

The authors state (lines 244-248, page 93): “Since the increased lysosomal surface area is linked to increased mTORC1 activity, it is possible that TFEB-dependent lysosomal biogenesis is an additional driver of the hyperactive mTORC1 in TSC1/2 deficient cells”. As stated above, previous studies have shown that TFEB promotes mTORC1 hyperactivation via transcriptional

induction of RagC/D (PMID: 28619945, 30843872). Once again, it is very surprising that the authors did not discuss these studies and did not even consider this mechanism as a possible explanation for the TFEB-mediated mTORC1 hyperactivation observed in TSC. Also, there is not convincing experimental evidence supporting that the increased lysosomal surface is linked to mTORC1 activation.

Response: We removed from the discussion the sentence in which we hypothesized that increased lysosomal surface area is linked to increased mTORC1 activity. We added and discussed the suggested references:

PMID: 28619945 Transcriptional activation of RagD GTPase controls mTORC1 and promotes cancer growth. DiMalta, Ballabio, et al. Science, 2017.

PMID:30843872, Myeloid Folliculin balances mTOR activation to maintain innate immunity homeostasis. Li, Arany, et al. JCI Insight, 2019.

6. Putative FLCN and TSC2 cooperation.

The data supporting cooperation between FLCN and TSC2 are very weak. The data in Figure 2b do not show a clear increase in TFEB nuclear localization in TSC2/FLCN-depleted cells compared to cells depleted of FLCN alone. In addition, the amount of nuclear TFEB observed in TSC2-depleted cells in Figure 3b is very different from that observed in the same cells treated in the same way in Figure 2d. Thus, the quantification shown in Figure 3c performed with 3 images only (as stated in the methods) is not sufficient to draw any conclusions. In addition to this, the amount of TFEB nuclear translocation in each treatment is likely dependent on the knockdown efficiency of every single protein. Moreover, the authors claim that overexpression of FLCN “rescues” TFEB phosphorylation in TSC2-silenced cells but the data shown are not convincing since total levels of TFEB-GFP are higher in TSC2-silenced cells upon FLCN overexpression relative to TSC2-silenced cells transfected with myc-vector in Fig 3g.

Therefore, claiming that the two proteins cooperate in the modulation of TFEB without the use of double-ko cells and, above all, without providing a clear mechanism is merely speculative and is not supported by solid data. The authors should provide solid data and a clear mechanism showing how FLCN and TSC2 co-ordinately modulate TFEB activity. Alternatively, this part should be removed from the manuscript.

Response: Regarding the differences between Figures 2d and 3b could be explained by the different methodology: in Figure 2d, we used a live imaging approach, while in Figure 3b the cells were fixed and then stained with anti-GFP antibody.

Regarding the quantitation in Figure 3b, we repeated the experiment and analyzed at least 80 cells from each condition, confirming that double knockdown of *TSC2* and *FLCN* increases the nuclear/cytoplasmic ratio of TFEB to a greater extent than knockdown of *TSC2* or *FLCN* alone (Fig. 3b, c). To further solidify these data, we found, in a new experiment, that double knockdown of *TSC2* and *FLCN* increases *GNMB* promoter activity to a greater extent than single knockdown, in both HeLa and HeLa-TFEB-GFP cells (Fig. 3d).

Regarding the strength of the data from double knockdown cells, we have shown that double knockdown of *TSC2* and *FLCN* increases the nuclear localization of TFEB (Fig. 3b, c), *GNMB* promoter activity (Fig. 3d), TFEB phosphorylation at S211 (Fig. 3e), and lysosomal gene expression (Fig. 3g) to a greater extent than knockdown of either gene individually.

Regarding levels of TFEB in Fig. 3g (now Fig. 3h), we repeated the experiment and normalized the levels of phospho-TFEB to total TFEB-GFP levels. Consistent with our prior

finding, expression of FLCN increased the phosphorylation of TFEB in *TSC2*-deficient cells by ~2-fold for S211 and ~1.7-fold for S142 (Fig.3h, i).

7. TFEB as a driver of TSC phenotype.

The data obtained by silencing TFEB in allografts generated with TSC2^{-/-} MEFs (in Fig 2k) suggest that TFEB is an important mediator of TSC phenotype. However, to obtain conclusive evidence the authors should generate tissue-specific TSC2/TFEB double-ko mice. The authors should at least discuss this approach.

Response: We agree that future work with tissue specific *TSC2/TFEB* double KO mice is a priority and have added this to the Discussion: "Further work will be needed to determine if *Tfeb* inactivation can alleviate renal disease in genetically engineered mouse models of TSC, as has been recently demonstrated for BHD-associated renal disease."

Reviewer #3 (Remarks to the Author):

The paper by Alesi et al describe a novel role for the Tuberous Sclerosis Complex in the regulation of lysosomal biogenesis. The authors report that, in TSC2-null mice and cells, a striking upregulation of lysosomal gene expression correlates with constitutive nuclear translocation of the master regulator of lysosomal biogenesis, TFEB. The authors further show that TFEB dysregulation upon TSC loss resembles that caused by loss of the FLCN protein, a GTPase Activating Protein (GAP) for the RagC GTPase. Accordingly, forced expression of a GTP-locked RagC mutant rescues the lysosomal upregulation triggered by TSC loss. Based on these data and the similarities with FLCN loss, the authors conclude that constitutive upregulation of lysosomal biogenesis through TFEB dysregulation could be a novel driver mechanism in Tuberous Sclerosis.

The manuscript is potentially interesting as it reveals a novel facet of the relationship between the mTORC1 pathway and lysosomal biogenesis, with potential relevance to tumorigenesis. However, the manuscript suffers from conceptual weaknesses and lack of mechanism in some key points. In particular, the claim that regulation of TFEB by TSC is mTORC1-independent is unsupported mechanistically and is logically inconsistent with the authors' own experiments.

Specific comments:

1. *Given the largely overlapping regulatory mechanisms between TFEB and TFE3 (i.e., PMID: 24448649, The nutrient-responsive transcription factor TFE3 promotes autophagy, lysosomal biogenesis, and clearance of cellular debris, Martina, Puertollano, et al. Sci Signal, 2014), it is unclear why only TFEB and not TFE3 is constitutively nuclear in TSC KO cells.*

Response: In new experiments, we show that, similarly to TFEB, endogenous TFE3 is nuclear in *Tsc1*-null MEFs in comparison to *Tsc1*-expressing MEFs (Supplementary Fig. 2b), in HeLa cells after *TSC2* knockdown by siRNA (Fig. 2c), in HeLa cells after *TSC2* knockout by CRISPR (Supplementary Fig. 3f), and in HEK293 cells after *TSC2* knockdown (Supplementary Fig. 3c). We have also added the cited reference about TFE3.

2. *Although upregulation of TFEB target genes in TSC cells correlates with increased TFEB localization in the nucleus, the two observations should be causally linked by showing that TFEB knockdown in TSC cells reverses the upregulation of lysosomal genes, both by qPCR and using the GPMNB luciferase assay.*

Response: In new experiments, we show that *TFEB* knockdown in *TSC2*-deficient cells decreases *GPNMB* promoter activity, *GPNMB* protein expression, and lysosomal gene expression (Supplementary Fig. 5a, b, c).

3. *By contrasting the coherent effects of TSC2 and FLCN deletion toward TFEB with their opposing roles in general mTORC1 signaling, the authors conclude that FLCN and TSC2 act on TFEB via mechanisms that do not involve mTORC1. This statement is problematic because it is not backed up by any mechanistic evidence. For example, does a phosphor-null TFEB specifically in the mTORC1 sites (S211A, S142A) fail to correct the upregulation of lysosomal target genes caused by TSC or FLCN loss?*

Response: As suggested, in new experiments we transfected HeLa cells with S142A TFEB, S211A TFEB and the double mutant (S142A/S211A). For all three mutants, TFEB localized predominantly to the nucleus in both control and *TSC2* knockdown cells (Supplementary Fig. 4).

4. *Related to the previous point, if expressing RagC-CA in TSC cells increases TFEB phosphorylation and inhibits TFEB target genes, isn't this evidence that regulation of TFEB by TSC is in fact mTORC1-dependent? Or are the authors proposing that S211 is phosphorylated by another protein kinase and, if so, which one?*

Response: Regarding the issue of mTORC1-dependency, the term “mTORC1 independent” has been replaced by “non-canonical” throughout the manuscript. We are not proposing that another kinase phosphorylates S211.

In closing, we thank the Editor and the Reviewers for their comments, which have helped us strengthen this manuscript.

REVIEWERS' COMMENTS

Reviewer #1 (Remarks to the Author):

The authors addressed my previous issues.

Reviewer #2 (Remarks to the Author):

The authors have satisfactorily addressed all of my criticisms except for point 6 "Putative FLCN and TSC2 cooperation". In my opinion, the data relative to this point are weak as most data were obtained by siRNA-mediated silencing and KO data are missing. Furthermore, the nature of such "coordinative regulation" of TFEB by FLCN and TSC is unclear. The authors' data just suggest that similarly to FLCN, TSC regulates RagC/D activity (either directly or indirectly).

Reviewer #3 (Remarks to the Author):

The authors have satisfactorily addressed my previous concerns and, also thanks to Reviewer 2's comments, clarified the proposed mechanism of TFEB regulation downstream of TSC loss in a manner that is much more logical, grounded in the data and consistent with the current literature. I have no further concerns.

Elizabeth Petri Henske, M.D.
Director, Center for LAM Research and Clinical Care
Pulmonary and Critical Care Medicine Division
Genetics Division (secondary appointment)
Brigham and Women's Hospital
Medical Oncologist, Genitourinary Oncology,
Dana-Farber Cancer Institute

Professor of Medicine,
Harvard Medical School

Associate Member,
The Broad Institute of MIT and Harvard

May 27, 2021

Dear Dr. Parish,

We are grateful for the positive feedback on the suitability of our manuscript entitled "TSC2 regulates lysosome biogenesis via a non-canonical RAGC and TFEB-dependent mechanism" (NCOMMS-20-41215A) for publication in Nature Communications. We appreciate the opportunity to address the remaining Reviewer 2 comment and resubmit the final manuscript.

Below, we address the specific comment of the reviewer 2.

Reviewer #2

1. *The authors have satisfactorily addressed all of my criticisms except for point 6 "Putative FLCN and TSC2 cooperation". In my opinion, the data relative to this point are weak as most data were obtained by siRNA-mediated silencing and KO data are missing. Furthermore, the nature of such "coordinative regulation" of TFEB by FLCN and TSC is unclear. The authors' data just suggest that similarly to FLCN, TSC regulates RagC/D activity (either directly or indirectly).*

Response: We agree that the nature of the coordinate regulation of TFEB by FLCN and TSC could be because TSC regulates RagC/D activity. We also agree that KO data would be interesting

to compare to the siRNA silencing, and updated the Results section with a following sentence:
“Further work using CRISPR mediated knockout of TSC2 and FLCN will be important to complement these siRNA-based findings”.

In closing, we thank the Editor and the Reviewers again for their comments, which have helped us strengthen this manuscript.

Sincerely,

Elizabeth P. Henske, M.D.